# Unreprogrammed H3K9me3 prevents minor zygotic genome activation and lineage commitment in SCNT embryos

Ruimin Xu[1,2,3,6], Qianshu Zhu [3,4,6], Yuyan Zhao[1], Mo Chen[2,5], Lingyue Yang[1,2], Shijun Shen[3,4], Guang Yang[3,4], Zhifei Shi[1], Xiaolei Zhang[1], Qi Shi[1,2], Xiaochen Kou[2], Yanhong Zhao[2], Hong Wang[2], Cizhong Jiang [3,4] ✉, Chong Li [2,3] ✉, Shaorong Gao [1,2,3] ✉ & Xiaoyu Liu [1,3] ✉

Somatic cell nuclear transfer (SCNT) can be used to reprogram differentiated somatic cells to a totipotent state but has poor efficiency in supporting full-term development. H3K9me3 is considered to be an epigenetic barrier to zygotic genomic activation in 2-cell SCNT embryos. However, the mechanism underlying the failure of H3K9me3 reprogramming during SCNT embryo development remains elusive. Here, we perform genome-wide profiling of H3K9me3 in cumulus cell-derived SCNT embryos. We find redundant H3K9me3 marks are closely related to defective minor zygotic genome activation. Moreover, SCNT blastocysts show severely indistinct lineage-specific H3K9me3 deposition. We identify MAX and MCRS1 as potential H3K9me3-related transcription factors and are essential for early embryogenesis. Overexpression of *Max* and *Mcrs1* significantly benefits SCNT embryo development. Notably, MCRS1 partially rescues lineage-specific H3K9me3 allocation, and further improves the efficiency of full-term development. Importantly, our data confirm the conservation of deficient H3K9me3 differentiation in Sertoli cell-derived SCNT embryos, which may be regulated by alternative mechanisms.

Somatic cell nuclear transfer (SCNT) refers to the transfer of a terminally differentiated somatic cell into an enucleated oocyte, which can reprogram the reconstructed embryo to a pluripotent state[1,2]. SCNT is of great value in animal resurrection and has vast prospective use in the treatment of human diseases[3–7]. To date, however, the development of cloned embryos is seriously defective, with undesired effects such as abnormal extraembryonic tissues and postnatal deficiency[8,9]. A large amount of evidence shows that a contributory cause of SCNT inefficiency lies in the donor cell-specific epigenome, which is not completely reprogrammed, resulting in a very large divergence from the fertilized embryos. These epigenetic memories include but are not limited to DNA methylation, histone modifications and 3D chromatin structures[9–17].

[1]Institute for Regenerative Medicine, Shanghai East Hospital, Shanghai Key Laboratory of Signaling and Disease Research, School of Life Sciences and Technology, Tongji University, 200120 Shanghai, China. [2]Shanghai Key Laboratory of Maternal Fetal Medicine, Shanghai Institute of Maternal-Fetal Medicine and Gynecologic Oncology, Clinical and Translation Research Center, Shanghai First Maternity and Infant Hospital, School of Life Science and Technology, Tongji University, 200092 Shanghai, China. [3]Frontier Science Center for Stem Cell Research, Tongji University, 200092 Shanghai, China. [4]Key Laboratory of Spine and Spinal Cord Injury Repair and Regeneration of Ministry of Education, Orthopaedic Department of Tongji Hospital, School of Life Sciences and Technology, Tongji University, 200092 Shanghai, China. [5]Chongqing Key Laboratory of Human Embryo Engineering, Center for Reproductive Medicine, Women and Children's Hospital of Chongqing Medical University, 400013 Chongqing, China. [6]These authors contributed equally: Ruimin Xu, Qianshu Zhu. ✉e-mail: czjiang@tongji.edu.cn; lichong@tongji.edu.cn; gaoshaorong@tongji.edu.cn; liuxiaoyu@tongji.edu.cn

H3K9me3 is generally considered to be one of the hallmarks of heterochromatin regions and plays important roles in gene regulation and transposon silencing[18–22]. H3K9me3-mediated heterochromatin remodeling is critical for natural reprogramming after fertilization, since either impairing or enforcing precocious acquisition of H3K9me3 results in compromised development[23–25]. During early embryonic development in mice, H3K9me3 exhibits distinct dynamic features in promoter and long terminal repeat (LTR) regions, and the precise deposition of H3K9me3 is pivotal to lineage specification[22,26,27].

H3K9me3-mediated heterochromatin is also regarded as a barrier to cell fate changes[18,28–30]. A previous study showed that H3K9me3 in donor cell genomes is a major obstacle to somatic cell reprogramming and that incomplete removal of H3K9me3 may lead to abnormal zygotic genome activation (ZGA) in 2-cell SCNT embryos[12,31]. Reducing H3K9me3 can restore the activation of H3K9me3-blocked regions and significantly improve SCNT embryo development[12,32–34]. Moreover, donor-inherited H3K9me3 impedes the removal of topologically associated domains (TADs) during SCNT embryo development, while overexpressing *Kdm4b*, an H3K9me3 demethylase, partially ameliorates abnormal 3D chromatin structures[15]. These results suggest that H3K9me3 is resistant to cellular transformation and thus may reduce the efficiency of reprogramming and the quality of SCNT embryos.

However, it is unclear how H3K9me3 is reprogrammed during the early development of SCNT embryos. Furthermore, trophectoderm (TE) separated from *Kdm4b*-overexpressing SCNT blastocysts showed abnormal gene expression levels, much more similar to those of the fertilized inner cell mass (ICM), suggesting that indiscriminate depletion of H3K9me3 might interfere with the lineage-specific deposition of H3K9me3 in SCNT blastocysts. In this study, we charted the H3K9me3 modification landscapes during the preimplantation development of mouse SCNT embryos. Comparison of data from fertilized embryos identified largely defective H3K9me3 reprogramming during SCNT embryogenesis. We found that persistent occupancy of H3K9me3 largely blocked the activation of minor ZGA genes and that lineage-specific H3K9me3 deposition was not evident in SCNT embryos. Our study not only reveals the molecular mechanisms underlying the failure of SCNT-mediated reprogramming but also identifies critical regulators of lineage-specific H3K9me3 deposition.

## Results

### Genome-wide excess H3K9me3 modification is present throughout the preimplantation development of SCNT embryos

To investigate the dynamics of H3K9me3 modification during early development of SCNT embryos, we collected cumulus cells (CCs) and SCNT embryos from the 1-cell (1C) stage (6 h post activation, 6 hpa) to the embryonic day 4 (E4) blastocyst stage and performed ultra-low-input chromatin immunoprecipitation followed by sequencing (ULI-NChIP-seq) (Fig. 1a)[26,35,36]. Embryos at the blastocyst stage were separated into the ICM and TE utilizing the micromanipulation technique. The replicates of H3K9me3 at each embryonic stage showed Pearson correlation coefficients > 0.90, indicating the high reproducibility and quality of our ChIP-seq data (Supplementary Fig. 1a, b).

As expected, H3K9me3 modification in SCNT embryos was highly enriched in LTR and long interspersed nuclear element (LINE) regions, in accordance with the pattern in fertilized embryos (Supplementary Fig. 1c, d)[26]. Notably, we observed consistently more H3K9me3-occupied regions in SCNT embryos at various stages than in fertilized embryos (Fig. 1b, c). These marks may block not only ZGA but also reprogramming at later stages[12]. *Zscan4d*, *Dux* and MT2/MERVL, for example, are important totipotency markers that are activated at the 2-cell stage in fertilized embryos but repressed in SCNT embryos[12,37–41]. The persistent H3K9me3 marks maintained from the donor cells to the 2-cell stage may be responsible for the silenced state of these regions in SCNT embryos because these regions in fertilized embryos showed almost no H3K9me3 signal (Fig. 1b). In general, the H3K9me3 signals in SCNT embryos were largely distinguished from those in fertilized embryos, suggesting a unique SCNT-mediated reprogramming process (Fig. 1d, e). Moreover, the reprogramming of H3K9me3 is highly dynamic during the preimplantation development of SCNT embryos, with drastic removal and deposition of H3K9me3 (Fig. 1e). Notably, H3K9me3 marks from the donor cells (CC) were gradually removed before the 8-cell stage, but the excessive H3K9me3 regions before ZGA in SCNT embryos may have significant effects (Fig. 1f). The dynamic trend of the CC-overlapped H3K9me3 in SCNT and fertilized embryos converged after the 2-cell stage, which was promptly restored at the morula and blastocyst stages, probably as a window of priming for further lineage segregation and development (Fig. 1f).

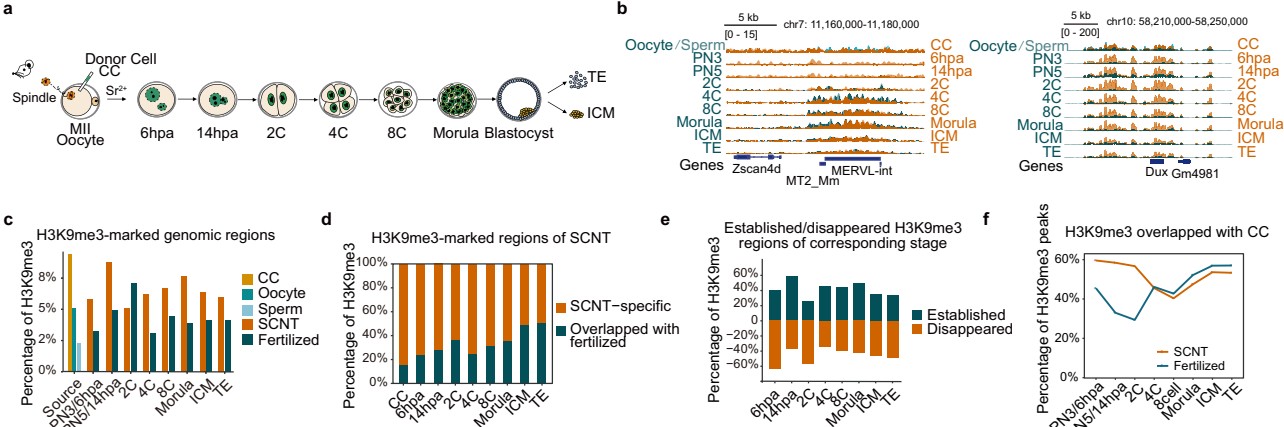

**Fig. 1 | Genome-wide excess H3K9me3 modification throughout SCNT pre-implantation development. a** Schematic of the preparation of mouse somatic cell nuclear transfer (SCNT) embryos for genome-wide ChIP-seq analysis of H3K9me3. Samples of donor cells (cumulus cells, CC) and 6 h post activation (hpa), 14 hpa, 2-cell (2C), 4-cell (4C), 8-cell (8C) and morula-stage embryos and the inner cell mass (ICM) and trophectoderm (TE) of E4 blastocysts were analyzed. **b** Genome Browser view of RPKM value of H3K9me3 signals around the *Zscan4d*, *Dux* and MERVL loci in SCNT (orange) and fertilized (cyan) embryos. **c** The percentage of genomic regions covered by H3K9me3 peaks in SCNT and fertilized embryos. **d** The percentage of H3K9me3 peaks in SCNT embryos overlapping with the corresponding stage of fertilized embryos. **e** The fractions of established and removed H3K9me3 domains during SCNT embryo development. **f** The percentage of H3K9me3 in SCNT and fertilized embryos overlapping with that in donor cells (CC) during preimplantation development. Source data are provided as a Source Data file.

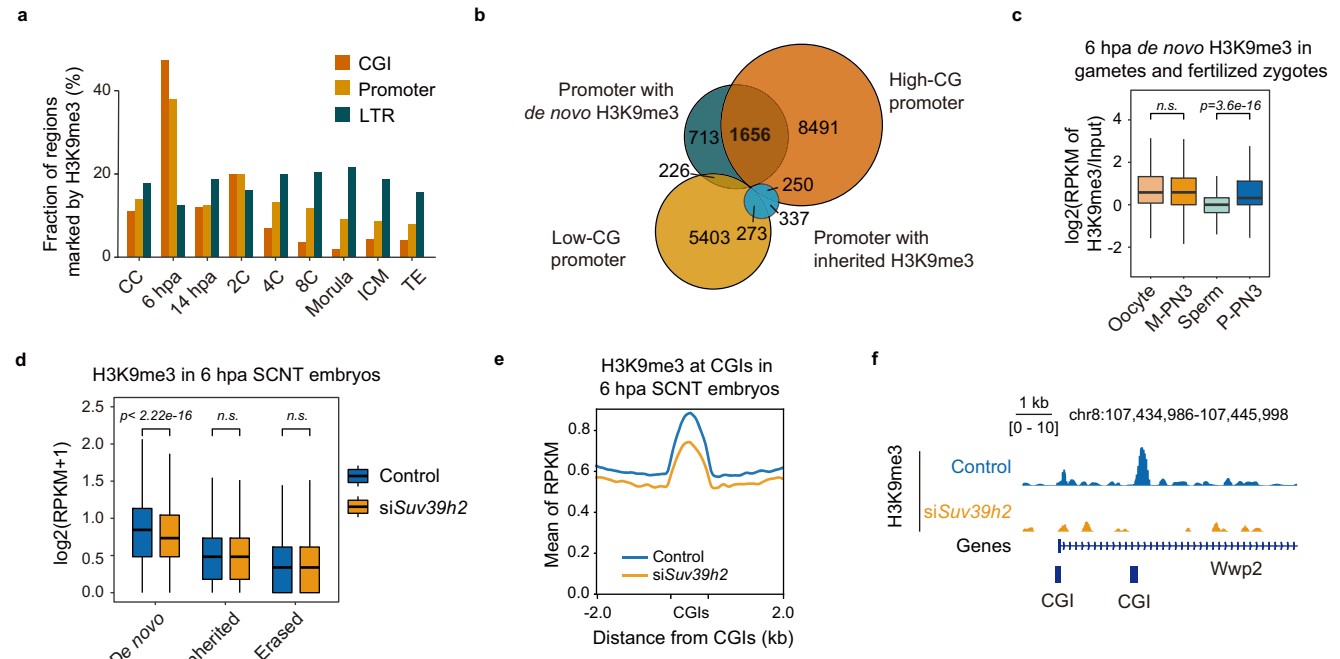

**Fig. 2 | CG-biased de novo H3K9me3 establishment appears shortly after SCNT activation. a** The fractions of CpG island, promoter and LTR regions marked by H3K9me3 peaks. **b** The overlap of promoters with de novo H3K9me3, inherited H3K9me3 from CCs, high CpG density and low CpG density at 6 hpa. **c** The H3K9me3 signal in oocytes, maternal pronuclei of fertilized zygotes, sperm and paternal pronuclei of fertilized zygotes in de novo H3K9me3 regions at 6 hpa. **d** The H3K9me3 signals in de novo, inherited and erased H3K9me3 regions in control and si*Suv39h2* SCNT embryos at 6 hpa. **e** H3K9me3 signals around CpG-islands in si*Suv39h2* and control SCNT embryos at 6 hpa. **f** H3K9me3 signals around the *Wwp2* gene locus in si*Suv39h2* and control SCNT embryos at 6 hpa (**c**). Statistical significance was calculated with a two-tailed Wilcoxon rank-sum test (**d**). Statistical significance was calculated with a two-tailed Student's *t* test. ns, no significance. Source data are provided as a Source Data file.

## Global loss of H3K9me3 occurs shortly after SCNT activation, accompanied by CpG-biased H3K9me3 deposition

In addition to the overall profile of H3K9me3 reprogramming during early SCNT embryogenesis, we were interested in the precise details of H3K9me3 remodeling, the underlying mechanisms and the difference compared with fertilized embryos.

It is worth mentioning that large-scale loss of H3K9me3 was observed in donor cells shortly after activation (Fig. 1c, e), with partial restoration at 14 hpa. Immunofluorescence staining of H3K9me3 in SCNT embryos from 5 mpi (minutes post-injection of donor cells) to 14 hpa showed the same trend (Supplementary Fig. 2a, b). Since PN3 is regarded as the stage of S-phase entry, we sought to determine whether DNA replication is responsible for this mass attenuation of H3K9me3[42]. Thus, at 0 hpa, we treated SCNT embryos for 6 h with aphidicolin, an inhibitor of DNA replication. We found no significant change in the H3K9me3 intensity, which suggested a replication-independent mechanism underlying this quick and large-scale loss of H3K9me3 (Supplementary Fig. 2c, d).

Interestingly, in addition to the global loss of H3K9me3 at 6 hpa, we also observed abundant enrichment of de novo H3K9me3 in CpG islands (CGIs) and promoters (Fig. 2a and Supplementary Fig. 2e). We defined de novo, inherited, and erased H3K9me3 based on the changes from CC to 6 hpa, and further found that most of the promoters with de novo H3K9me3 (1656/2595, ~65%) also possessed a high CpG content, in contrast to the low percentage of de novo H3K9me3 in promoters possessing a low CpG content (226/5902, ~3.8%), while no preference for CpG content was observed in promoters with CC-inherited H3K9me3 (Fig. 2b). We then sought to determine the significance of such CpG-biased behavior. Surprisingly, regions with de novo H3K9me3 in 6 hpa SCNT embryos also showed H3K9me3 deposition in the paternal pronuclei of PN3 zygotes but not in the corresponding maternal pronuclei (Fig. 2c). Notably, the de novo

H3K9me3 in 6 hpa SCNT embryos was also consistent with the enrichment of H3K9me3 in CGIs in oocytes (Supplementary Fig. 2f). Moreover, the genes marked by H3K9me3 in MII oocytes and by de novo H3K9me3 in 6 hpa SCNT embryos were both enriched in gene ontology (GO) terms related to organ morphogenesis (Supplementary Fig. 2g). De novo H3K9me3 tends to suppress premature expression, priming genomic regions for chromatin compaction at later developmental stages[24,26]. These results indicate that the reprogramming of H3K9me3 in SCNT embryos tends to simulate the developmental state of fertilized embryos.

Inspired by a previous study suggesting that SUV39H2 catalyzes de novo H3K9me3 in the paternal pronucleus after fertilization[24], we knocked down *Suv39h2* in SCNT embryos by siRNA injection (Supplementary Table 1). As expected, the abundance of de novo H3K9me3 at 6 hpa was significantly reduced, while the inherited and erased H3K9me3 were unaffected (Fig. 2d). Furthermore, H3K9me3 at CGIs in 6 hpa SCNT embryos was dramatically diminished after *Suv39h2* knockdown (Fig. 2e, f). In conclusion, H3K9me3 exhibits genome-wide loss, accompanied by CpG-biased de novo deposition shortly after activation. SUV39H2 catalyzes the de novo deposition of H3K9me3 at 6 hpa, completing the transition of donor cells from a somatic state to an intermediate gamete-like state and finally to a zygote-like state in SCNT embryos, which attempts to simulate the chromatin state of fertilized embryos.

## Activation of minor ZGA genes is blocked by CC-inherited H3K9me3

H3K9me3 modifications in somatic cells are considered to be a barrier to cell fate changes and cover the regions that failed to be activated in 2-cell SCNT embryos[12], Meanwhile, the establishment of H3K9me3 in early embryos are also of significant importance[26]. These emphasize the importance of revealing how the H3K9me3 modifications in donor

cells were removed or inherited before ZGA. To address this uncertainty, we classified H3K9me3 regions based on whether they could be erased until 14 hpa, when minor ZGA occurred (Supplementary Fig. 3a, b). H3K9me3 removed at 6 hpa or 14 hpa was defined as "reprogrammed H3K9me3 (rpgH3K9me3)" (6 hpa-rpg and 14 hpa-rpg), while H3K9me3 maintained at 14 hpa or 2-cell stage was defined as "unreprogrammed H3K9me3 (unrpgH3K9me3)" (14 hpa-unrpg and 2C-unrpg). H3K9me3-marked heterochromatin domains prevent the binding of diverse transcription factors (TFs) and are negatively correlated with chromatin accessibility[43]. As expected, chromatin accessibility in SCNT embryos exhibited a dramatically lower level in unrpgH3K9me3 regions, compared to those rpgH3K9me3 regions (Supplementary Fig. 3c)[44]. Interestingly, the binding motifs of chromatin structure-related TFs, such as CTCF, were more likely to be located in the rpgH3K9me3 regions (Fig. 3a). This might be because the boundaries of TADs in heterochromatin regions of CCs are much more easily affected by reprogramming factors. In contrast, the binding motifs of TFs related to enhancer-promoter interactions, including YY1, were more likely to be packaged in unrpgH3K9me3 regions, which might hinder the enhancer-dependent activation of ZGA genes (Fig. 3a)[44]. Furthermore, overexpression (OE) of *Kdm4b* can erase H3K9me3 signals in those regions that were unreprogrammed (Supplementary Fig. 3d, e and Supplementary Table 2).

To further understand how the H3K9me3 modifications in donor cells affect normal ZGA in SCNT embryos, we initially defined genes that were restrictedly, partially and fully activated during minor ZGA, major ZGA and MGA (mid-preimplantation gene activation; refers to the activation of genes at the 4-cell and subsequent stages), according to their expression levels compared to those in fertilized embryos (Supplementary Fig. 3f, g). Notably, only minor ZGA-restricted genes were significantly enriched by H3K9me3 from donor cells (Fig. 3b). Moreover, the CC H3K9me3 level showed a positive correlation with the degree of repression of minor ZGA genes in SCNT embryos, but was unrelated to the major ZGA and MGA genes (Supplementary Fig. 3h). It is worth mentioning that the majority of H3K9me3-marked regions in CC showed reduced H3K9me3 signal and were identified as reprogrammable H3K9me3 regions (Supplementary Fig. 3b), which suggested that these H3K9me3 modifications may not be strong enough to directly and completely block the transcription of the marked major ZGA genes. Next, we sorted the list of minor ZGA-restricted genes that were directly covered by persistent H3K9me3 in SCNT embryos (Fig. 3c). The affected genes included *Dux* and its target genes (such as *Zscan4* family genes), which have been proven to be important for preimplantation development (Fig. 3c and Supplementary Fig. 3i)[17,38,39,45]. Also included was a family of genes that encode deubiquitinating enzymes, namely, DUB1 (*Usp17la/b*), DUB2 (*Usp17lc/d*) and DUB3 (*Usp17le*). They are also DUX target genes and transiently expressed at the early 2-cell stage of fertilized embryos but completely repressed in SCNT embryos (Fig. 3d, e)[46–49]. Loss of DUB2 leads to failure of hatching and results in early embryonic lethality at E6.5, suggesting the important role of DUBs in preimplantation development[50]. Injection of *Usp17lc* or *Usp17ld* mRNA in SCNT embryos improved the efficiency of SCNT blastocyst formation, indicating that activation of H3K9me3-marked minor ZGA genes is necessary for further embryonic development (Fig. 3f and Supplementary Table 2). In addition, we found that only H3K9me3 signals at minor ZGA genes, especially the restricted ones at the late 1-cell stage, was decreased in response to *Kdm4b* OE (Fig. 3b–d). Given that persistent H3K9me3 attenuates minor ZGA, this might explain why SCNT embryos lacking *Kdm4b* expression undergo severe 2-cell arrest[34].

## UnrpgH3K9me3 hampers the timely activation of repeat elements in SCNT embryos

In addition to protein-coding genes, retrotransposons, including LINE, SINE and LTR, are also transiently activated during ZGA in fertilized embryos, which is triggered by drastic DNA demethylation and suppressed by H3K9me3 deposition[26,27,51,52]. Generally, SCNT embryos showed delayed activation of most retrotransposons, which was correlated with abnormal DNA re-methylation and/or unrpgH3K9me3 modification (Fig. 3g, h and Supplementary Fig. 3j–l)[10]. For example, MERVL (MERVL-int and MT2_Mm), as the marker of totipotency and dominantly transcribed at the L2C stage, was not highly activated until the 4C stage in SCNT embryos (Fig. 3g). Correspondingly, H3K9me3 was retained on MERVL in SCNT embryos before ZGA, indicating that unrpgH3K9me3 may hinder the timely expression of repeat elements (Fig. 3h). *Kdm4b* OE partially erased redundant H3K9me3, rather than diminished de novo H3K9me3, on MERVL regions during minor ZGA, which might further benefit major ZGA (Fig. 3i). Intriguingly, we found that the expression level of retrotransposons in L2C embryos exhibited stronger negative correlations with the H3K9me3 level in L1C embryos, suggesting that H3K9me3 deposition tends to interfere with transcription at the latter stage (Fig. 3j). Taken together, these results indicate that unrpgH3K9me3 inherited from donor cells is significantly correlated with the restricted minor ZGA genes and the delay of the activation of repeat elements in SCNT embryos.

## Indistinct lineage-specific H3K9me3 in SCNT embryos during the first cell fate determination

After several cell divisions, compacted blastomeres differentiate into the ICM and TE, a process referred to as the first cell fate determination. H3K9me3 modification plays a pivotal role in lineage commitment during embryonic development[22,26,28]. Our previous study showed that H3K9me3 is deposited at promoters in a lineage-specific manner after implantation to repress lineage-incompatible genes[26]. However, how H3K9me3 is differentially deposited during SCNT embryo development remains elusive. Surprisingly, we found that the difference of H3K9me3 between the ICM and TE of SCNT embryos was quite unnoticeable, although the fertilized ICM and TE were thoroughly separated from each other (Fig. 4a). Notably, SCNT blastocysts generated by using immature Sertoli cells as donor cells also exhibited the same deficiency (Supplementary Fig. 4a). We then identified lineage-specific H3K9me3-marked regions in fertilized blastocysts (Fig. 4b and Supplementary Fig. 4b, c). In contrast to the fertilized embryos, we observed a lack of lineage-specific H3K9me3 differentiation in SCNT embryos during the first cell fate determination, and this deficiency can be traced back to the morula stage (Fig. 4b, c and Supplementary Fig. 4c). To explore the reasons for such insufficient H3K9me3 differentiation in SCNT blastocysts, we annotated the lineage-specific regions with higher or lower levels of H3K9me3 in SCNT blastocysts, compared to the fertilized ones. Interestingly, regions with lower H3K9me3 signals in SCNT embryos were significantly enriched in promoter regions (Supplementary Fig. 4d). Thus, we proposed that certain TFs or chromatin regulators, which are deficient in SCNT embryos, may be responsible for this severe defect. To test this hypothesis, we examined the enrichment of lineage-specific H3K9me3 at TF binding sites based on public mESC ChIP-seq data in both ICM and TE of fertilized and SCNT embryos. Interestingly, compared with fertilized embryos, cumulus cell-derived SCNT embryos did not exhibit adequate deposition of H3K9me3 at the binding sites of most of the TFs and chromatin regulators in either ICM or TE (Fig. 4d and Supplementary Fig. 4e). In addition, TF binding sites in both types of SCNT blastocysts showed a much lower differential allocation of H3K9me3 between the ICM and TE (Supplementary Fig. 4e). The emerging factors that might contribute to the lineage-specific deposition of H3K9me3 included TRIM28 and SETDB1, which are typical H3K9me3-related factors (Supplementary Fig. 4f), consistent with our previous results based on motif enrichment analysis of fertilized mouse embryos[26,53,54].

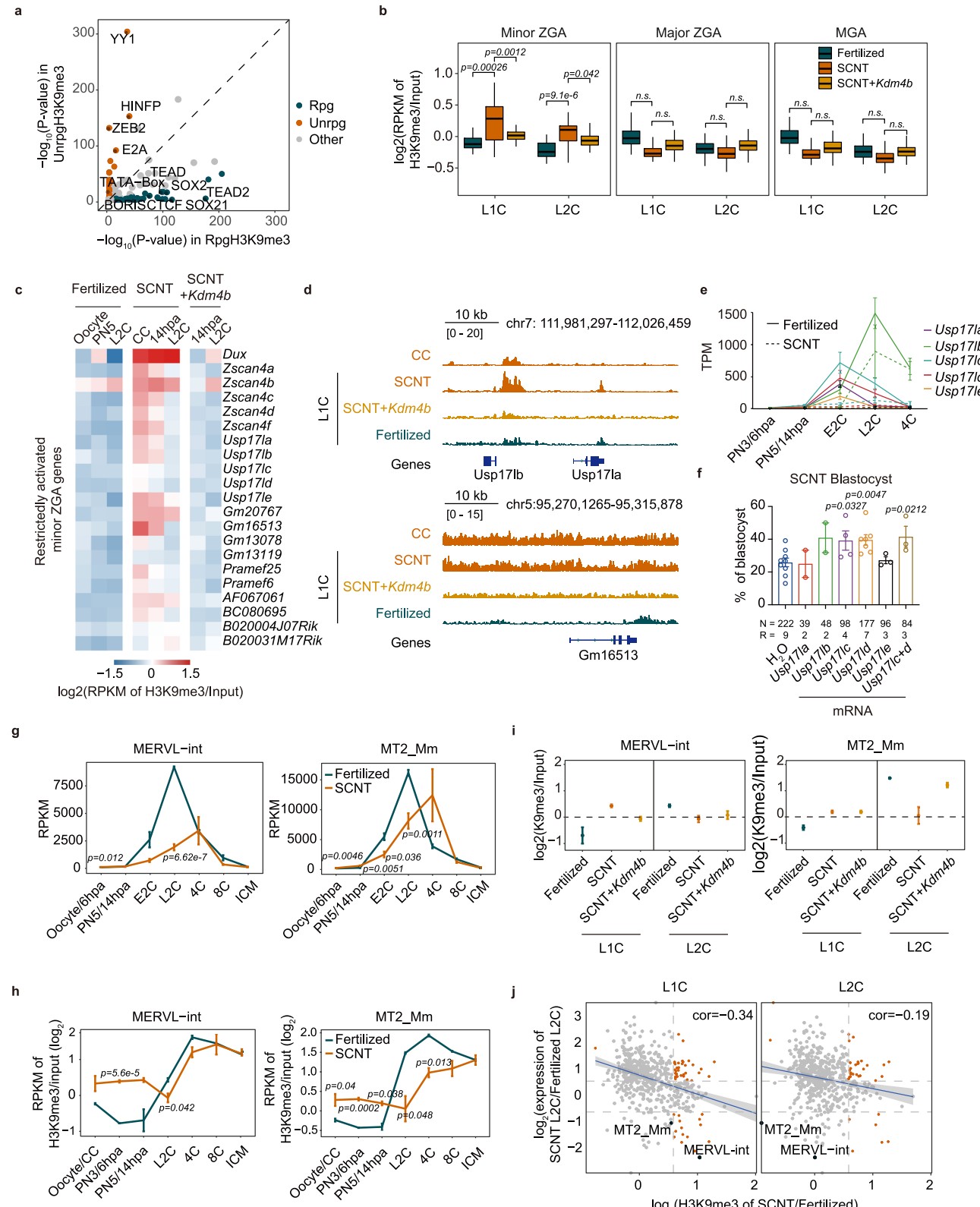

## Max and Mcrs1 are critical for preimplantation development and lineage-specific H3K9me3 deposition but are severely deficient in SCNT embryos

Furthermore, we selected other important TFs, including MAX, MCRS1 and MYCN (Supplementary Fig. 4f, g). Previous studies have shown that these factors are pivotal for embryo development, whose depletion cause embryonic lethality from E3.5 to E10.5[55–58]. MYC is involved

in the regulation of self-renewal and pluripotency of mESCs[59]. MAX is a partner of the MYC family and has been reported to interact with G9A and GLP (H3K9 methyltransferases) to repress germ cell-specific genes in ESCs[60]. MCRS1 is a subunit of the NSL complex and a putative regulator of the chromatin remodeling INO80 complex[61]. MCRS1 is also identified as a key factor for epiblast formation, inducing naïve pluripotency in hESCs[62]. These results indicate their roles in epigenetic

**Fig. 3 | Unreprogrammed H3K9me3 directly blocks minor ZGA and delays LTR activation. a** *P*-values (−log10 transformed) of representative gene motifs in rpgH3K9me3 (reprogrammed) and unrpgH3K9me3 (unreprogrammed) regions. **b** H3K9me3 signals on ZGA and MGA genes in late 1-cell and late 2-cell fertilized, SCNT and SCNT+*Kdm4b* embryos. **c** H3K9me3 signal on restrictedly activated minor ZGA genes in fertilized, SCNT and SCNT+*Kdm4b* embryos. **d** H3K9me3 signal around the *Usp17lb, Usp17la* and *Gm16513* loci in CC, SCNT, SCNT+*Kdm4b* and fertilized embryos at the late 1-cell stage. **e** Expression level of *Usp17l* family genes in fertilized and SCNT embryos. *n* = 6 (SCNT) and *n* = 2 (Fertilized, from published data[82] [GSE71434]) biologically independent samples. **f** The blastocyst rate of SCNT embryos after overexpression of *Usp17l* family genes. *N* indicates the total number of embryos. *R* indicates the number of replicates. **g** The mean expression level of MERVL-int and MT2_Mm in SCNT and fertilized embryos. *n* = 2 (Fertilized, from published data[82] [GSE71434]) and *n* = 6 (SCNT) biologically independent samples. **h** The dynamics of the mean H3K9me3 signal of MERVL-int and MT2_Mm in SCNT

and fertilized embryos. The number of biologically independent samples: *n* = 2 for SCNT-ICM and the fertilized group from a previous publication[26] (GSE97778); *n* = 4 for CC and SCNT-6 hpa; *n* = 3 for SCNT-14 hpa, L2C, 4C, and 8C. **i** The mean H3K9me3 signals of MERVL-int and MT2_Mm loci in fertilized, SCNT and SCNT +*Kdm4b* embryos at the late 1-cell and late 2-cell stages. The number of biologically independent samples: *n* = 2 for SCNT+*Kdm4b*-L1C and the fertilized group from a previous publication[26] (GSE97778); *n* = 3 for SCNT group and SCNT+*Kdm4b*-L2C. **j** The correlation of expression and H3K9me3 signal of LTRs in SCNT embryos compared with fertilized embryos. The Pearson correlation coefficient is shown. The red points indicate LTRs with significant changes in both H3K9me3 and expression levels. The gray points indicate other LTRs (**e–i**). Data are presented as mean values ± SEM (**b**). Statistical significance was calculated with a one-tailed Wilcoxon rank-sum test (**f–h**). Statistical significance was calculated with a two-tailed Student's *t* test. ns, no significance. Source data are provided as a Source Data file.

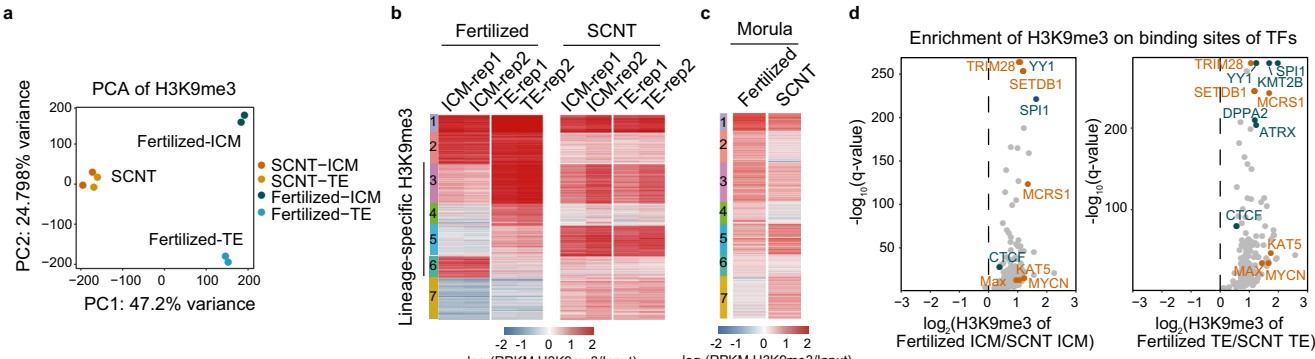

**Fig. 4 | Indistinct lineage-specific H3K9me3 in SCNT embryos during the first cell fate determination. a** PCA of H3K9me3 signals in the ICM and TE of fertilized and SCNT embryos. **b** H3K9me3 signals in the ICM and TE of fertilized and SCNT embryos; the regions with the most positive and negative loading (*n* = 3427, Supplementary Fig. 4b) were classified into 7 clusters based on H3K9me3 signals using the k-means function. **c** H3K9me3 signals in the morula of fertilized and SCNT embryos, with these regions corresponding to (**b**). **d** The differences in H3K9me3 levels at TF binding sites between fertilized and SCNT embryos; left panel: ICM, right panel: TE. The *Y*-axis shows the −log10 transformed *q*-values calculated using Fisher's exact test, and the *X*-axis shows the log2-fold change values. The red plots represent the potential TFs that may be related to defective differential H3K9me3 deposition in SCNT blastocysts. The blue plots represent well-known chromatin architecture-related and H3K9me3-related factors. The gray plots represent other TFs. Source data are provided as a Source Data file.

reprogramming during early lineage commitment. Notably, these factors lack H3K9me3 deposition at their binding sites and are repressed in cumulus cell-derived SCNT blastocysts (Supplementary Fig. 4f, g and 5a–c). We first sought to determine how fertilized embryos are affected in the lack of these factors. Notably, knockdown of *Max* or *Mcrs1* by morpholino (MO) injection into fertilized zygotes dramatically impeded preimplantation embryo development, with hardly any formation of E3.5 blastocysts, consistent with the developmental arrest of *Max*[-/-] and *Mcrs1*[-/-] embryos near the time of implantation observed in previous studies (Fig. 5a–c, Supplementary Fig. 5d and Supplementary Table 3)[55,56]. In contrast, knockdown of *Mycn* had little impact on blastocyst formation (Fig. 5b, c and Supplementary Fig. 5d). Furthermore, blastocysts lacking MAX and MCRS1 showed an abnormal morphology, with an unclear distinction between ICM and TE and decreased numbers of both Oct4+ and Cdx2+ cells (Fig. 5d and Supplementary Fig. 5e). We then asked how the absence of MAX or MCRS1 can generate such a severe phenotype. Transcriptome analysis of *Max* MO and *Mcrs1* MO blastocysts suggested that both the ICM and TE lineages differed significantly from those of the control MO blastocysts (Fig. 5e). Moreover, *Max* and *Mcrs1* MO influenced the expression of multiple genes encoding enzymes related to chromatin and histone modifications, such as *Suv39h1, Dnmt1, Ezh2, Eed* and *Kmt2c* (Fig. 5f)[63–67]. In summary, these results emphasize that MAX and MCRS1 play essential roles in early mouse embryo development, especially during the first cell fate decision, and may participate in the establishment of lineage-specific H3K9me3.

## Overexpression of *Max* or *Mcrs1* greatly benefits both pre- and post-implantation development of SCNT embryos

We then verified whether overexpression of *Max* or *Mcrs1* in SCNT embryos can improve SCNT-mediated reprogramming, especially during the first cell fate determination. To this end, *Max* or *Mcrs1* mRNA was injected into both blastomeres of SCNT embryos at the L2C stage (Fig. 6a and Supplementary Table 2). Either *Max* or *Mcrs1* OE significantly improved the preimplantation development of cumulus-derived SCNT embryos (Fig. 6b, c). In addition, *Max* and *Mcrs1* OE SCNT blastocysts contained increased numbers of both Oct4+ and Cdx2+ cells (Supplementary Fig. 6a, b). Both *Max* and *Mcrs1* OE blastocysts showed a more intact morphology and more distinct segregation of the ICM (Oct4+) and TE (Cdx2+) lineages (Supplementary Fig. 6a). Moreover, approximately 2–3-fold more E10.5 embryos were obtained after *Max* or *Mcrs1* OE (Fig. 6d, e and Supplementary Fig. 6c, d). These results suggest that MAX and MCRS1 indeed increased the quality of cumulus-derived SCNT blastocysts and enhanced their implantation capability. Unfortunately, *Max* OE seemed to have little effect on the birth rate (Supplementary Fig. 6e). Surprisingly, *Mcrs1* OE successfully increased the birth rate of SCNT embryos by approximately 4-fold (Fig. 6f). The full-term SCNT fetuses and placentas of *Mcrs1* OE embryos had a normal weight and exhibited healthy growth (Supplementary Fig. 6f–h). On the other side, *Mcrs1* OE had no obvious effect on Sertoli cell-derived cloning efficiency, which may be due to the sufficient expression of *Mcrs1* in 2-cell and 4-cell Sertoli cell-derived SCNT embryos (Supplementary Fig. 5c, 6i).

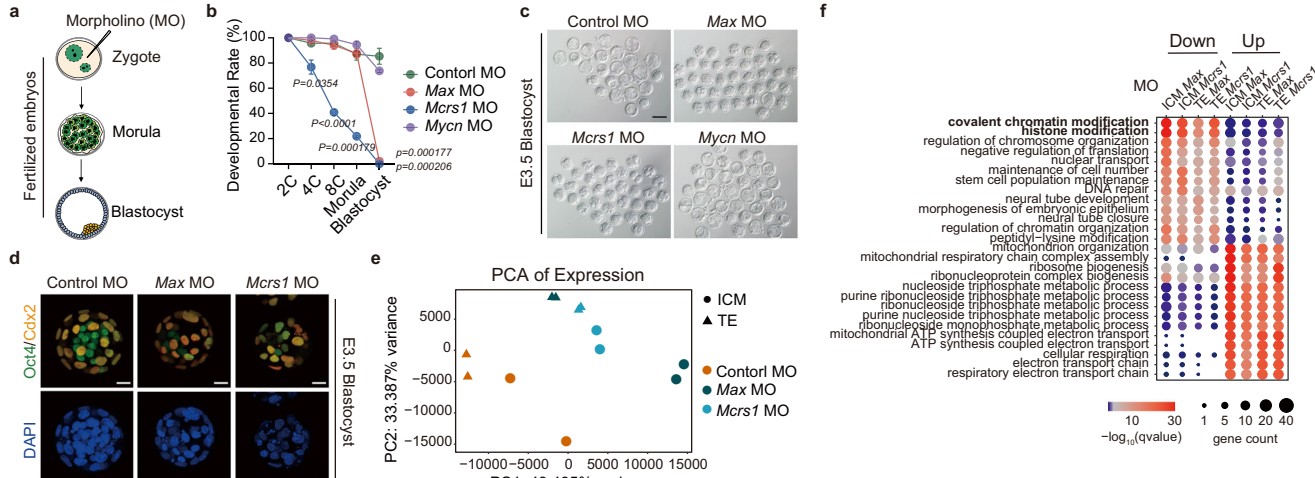

**Fig. 5 | *Max* and *Mcrs1* are pivotal for mouse preimplantation development and lineage-specific H3K9me3 deposition. a** Schematic of the functional study of *Max* and *Mcrs1* in fertilized embryos. **b** Growth curves of preimplantation embryo development after injection with 1 mM morpholinos targeting *Max* (*Max* MO), *Mcrs1* (*Mcrs1* MO), *Mycn* (*Mycn* MO) or a 25-base random control MO (control MO). Control MO (n = 23, 24, 22), *Max* MO (n = 28, 38, 36), *Mcrs1* MO (n = 22, 35, 35), *Mycn* MO (n = 21, 36, 37). **c** Representative images of *Max*, *Mcrs1*, and *Mycn* MO-injected embryos at embryonic day 3.5 (E3.5). Scale bar, 100 μm. **d** Immunofluorescence staining of Oct4 (green) and Cdx2 (yellow) in control MO, *Max* MO and *Mcrs1* MO embryos at E3.5. DNA was stained with DAPI (blue). Scale bar, 20 μm. 3 experiments were repeated independently with similar results. **e** Principal component analysis of the expression levels in replicates of the ICM and TE from fertilized embryos injected with *Max* MO (0.3 mM), *Mcrs1* MO (0.2 mM) and control MO (1 mM). Each sample had two biological replicates. **f** Gene ontology analysis of the genes with downregulated and upregulated expression in *Max* MO- and *Mcrs1* MO-injected samples. The color of the circles represents the −log10 transformed *q*-values, and the size indicates the number of genes included (**b**). Data are presented as mean values ± SEM. Statistical significance was calculated with a two-tailed Student's *t* test. Source data are provided as a Source Data file.

## Overexpression of *Mcrs1* partially recovers lineage-specific H3K9me3 allocation in SCNT embryos

We then asked how MCRS1 affected the deposition of lineage-specific H3K9me3 during the first cell fate decision. The overall H3K9me3 distributions in the ICM and TE of cumulus cell-derived SCNT blastocysts after *Mcrs1* OE showed higher similarity with those in fertilized blastocysts, both significantly distinct from those in control SCNT blastocysts (Fig. 6g). However, H3K9me3 in Sertoli cell-derived SCNT blastocysts after *Mcrs1* OE persisted close to the control ones (Supplementary Fig. 6j). In detail, nearly 33% of lineage-specific H3K9me3 regions were rescued in cumulus cell-derived SCNT embryos after *Mcrs1* OE (Supplementary Fig. 6k). Importantly, MCRS1 positively regulated H3K9me3 on multiple genes encoding histone modification modifiers, such as *Trim28, Dnmt3a, Kmt2c*, and *Ezh2*, which corresponded to our result of *Mcrs1* MO in fertilized embryos (Fig. 5f and Supplementary Fig. 6l). These findings suggested that MCRS1 may regulate lineage-specific H3K9me3 in an indirect manner, by affecting genes related to lineage commitment. We further examined the changes of H3K9me3 levels at MCRS1 binding sites. *Mcrs1* OE led to strengthened H3K9me3 signals on MCRS1 target regions in both lineages (Fig. 6h, i and Supplementary Fig. 6m). Lineage specificity of H3K9me3 in ICM and TE was also enhanced, indicating that MCRS1 may also directly induce H3K9me3 deposition (Supplementary Fig. 6 m, n). Remarkably, *Mcrs1* OE failed to rescue the lineage-specific H3K9me3 deposition on certain developmental genes, which are critical for lineage commitment (eg. *Gata3, Gata6, Fgfr1*) (Supplementary Fig. 6k, l). This indicates the existence of other critical factors that need to be validated. Notably, *Mcrs1* OE in Sertoli cell-derived SCNT blastocysts dramatically enlarged the lineage differentiation of H3K9me3 deposition on various TF binding sites, yet led to a direction opposite that of the fertilized ones (Supplementary Fig. 6o). This indicated that moderate expression of *Mcrs1* is essential for the proper deposition of H3K9me3. Taken together, our data suggest that *Mcrs1* OE can partially correct the abnormal H3K9me3 in cumulus cell-derived SCNT embryos by guiding lineage-specific H3K9me3 distribution on epigenetic regulators or transcription factors that are critical for future lineage differentiation and embryo development.

## Discussion

Nearly thirty years have passed since the first cloned mammalian animal was born, and the first nonhuman primate was also successfully cloned recently[4,6]. SCNT provides us with a powerful tool to study the mechanism of cell fate transition from the differentiated to the pluripotent state, as well as promising application prospects in human therapeutics[7]. However, the rather poor efficiency of full-term development (-1–2% in mice) greatly limits the application of SCNT[68,69]. H3K9me3-marked heterochromatin in donor cells has been proven to be an obstacle for SCNT reprogramming in mammalian species, impeding genome activation in 2C embryos. SCNT embryos with the removal of H3K9me3 by *Kdm4b/d* overexpression or *Suv39h1/2* knockdown showed dramatically improved preimplantation development by overcoming 2-cell arrest, although the final birth rate is still much lower than that of fertilized embryos[12,34]. This emphasizes the importance of precise regulation of H3K9me3 deposition rather than removal at later stages, probably during the first lineage differentiation. In this study, we examined genome-wide H3K9me3 modification during the preimplantation development of SCNT embryos and found that SCNT blastocysts are severely defective in lineage-biased H3K9me3 deposition. Integrated analysis of transcriptome, H3K9me3 enrichment and public ChIP-seq data from mESCs identified multiple transcription factors and chromatin regulators that exhibited failure of lineage-specific H3K9me3 deposition at their binding sites and were transcriptionally repressed in SCNT embryos, including TRIM28, SETDB1, YY1, SPI1, KAT5, MAX, MCRS1, and MYCN. Our data provided numerous potential factors for regulating cell-type-specific H3K9me3 allocation, which provided a new perspective for restoring the efficiency of SCNT based on TF-regulated lineage-specific epigenomic differentiation. On the other hand, the examination of H3K9me3 in SCNT embryos is still a mixed result regardless of the developmental capacity. Singe-cell-based ChIP-seq technology will largely help us focus only on the samples showing unreprogrammed

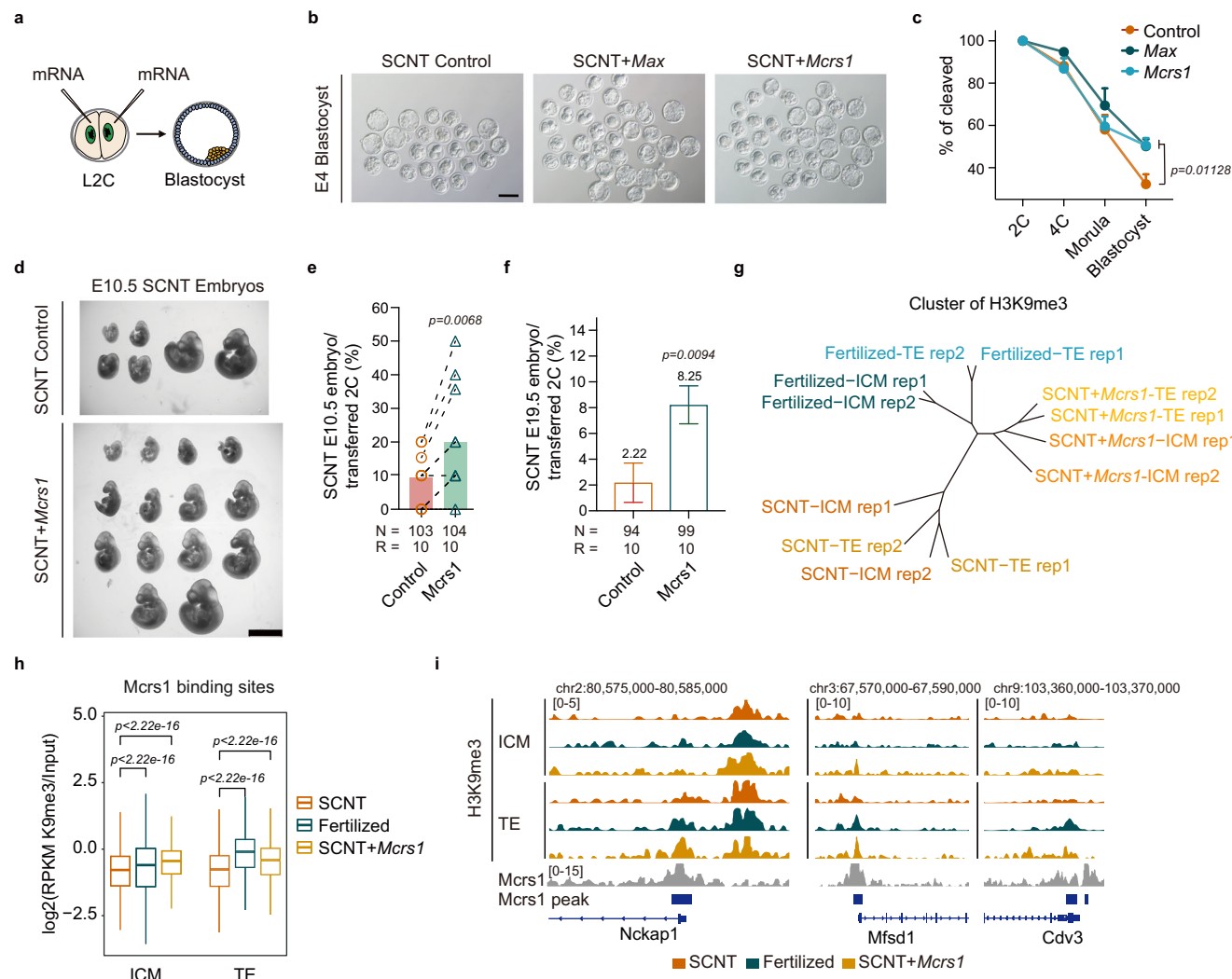

**Fig. 6 | Overexpression of *Max* or *Mcrs1* significantly improves cumulus cell-derived SCNT development. a** Schematic of the strategy for overexpression of *Max* and *Mcrs1* in SCNT embryos. Control (H$_2$O), *Max* (50–100 ng/µL) and *Mcrs1* (250–500 ng/µL) mRNA were injected into both blastomeres of late 2-cell embryos. **b** Representative images of *Max* and *Mcrs1* OE embryos at E4. Scale bar, 100 µm. **c** Overexpression of *Max* or *Mcrs1* significantly improved the blastocyst rate of SCNT embryos. Control, 226 embryos, *r* = 7; *Max*, 240 embryos, *r* = 6; *Mcrs1*, 171 embryos, *r* = 5. *r* indicates the number of replicates. **d** Representative images of E10.5 *Mcrs1* OE SCNT embryos. One representative result of at least three independent experiments is shown. Scale bar, 2 mm. **e** *Mcrs1* OE greatly improved the implantation ability of SCNT embryos. *N* indicates the number of transferred 2-cell embryos. *R* indicates the number of transferred pseudopregnant female mice.

Spots connected by a dotted line represent the same pseudopregnant female mouse. **f** The birth rate of full-term SCNT pups after *Mcrs1* OE. *N* indicates the number of transferred 2-cell embryos. *R* indicates the number of transferred pseudopregnant female mice. **g** Unrooted dendrogram of H3K9me3 signals between the ICM and TE from fertilized, SCNT and SCNT+*Mcrs1* embryos mediated from hierarchical cluster analysis. **h** H3K9me3 signals in fertilized, SCNT and SCNT+*Mcrs1* blastocysts on the binding sites of MCRS1. **i** H3K9me3 signals in ICM and TE of fertilized, SCNT and SCNT+*Mcrs1* embryos, and MCRS1 binding signals (published data from GSE51746) around the *Nckap1*, *Mfsd1*, and *Cdv3* loci (**c**, **f**). Data are presented as mean values ± SEM (**c**, **e**, **f**). Statistical significance was calculated with a two-tailed Student's *t* test (**h**). Statistical significance was calculated with a two-tailed Wilcoxon rank-sum test. Source data are provided as a Source Data file.

epigenomes. Moreover, given that the deficient lineage-specific H3K9me3 deposition may already appear in SCNT morulae (or maybe even earlier), the single-blastomere investigation can greatly empower us to confirm such speculation. Moreover, multi-omic single-cell technologies will further improve our understanding of the regulatory mechanisms of the potential TFs related to lineage-specific H3K9me3 establishment.

A previous study emphasized the influence of H3K9me3 on ZGA[12]. For further analysis, we generated more detailed H3K9me3 ChIP-seq data, especially at time points before major ZGA (CC, 6 hpa, 14 hpa, 2C stage). Based on the dynamic pattern of H3K9me3, the barrier was pushed forward to minor ZGA. Although only a tiny fraction of H3K9me3-marked regions inherited from CCs were completely removed at 14 hpa (L1C stage), most of the residual H3K9me3 marks

showed reduced signals and were largely defined as reprogrammable regions at the L2C stage (Supplementary Fig. 3a, b). Correspondingly, neither major ZGA nor MGA genes possessed stronger H3K9me3 signals in SCNT embryos than in fertilized embryos, and *Kdm4b* OE eliminated excessive H3K9me3 only on minor ZGA genes (Fig. 3b). These results further clarified how H3K9me3 can directly hinder ZGA in SCNT embryos.

We found a global loss of H3K9me3 signals after activation that was replication-independent. However, the mechanism is still unclear. It is remarkable that near-exclusive expression of *Kdm4a*, a lysine-specific demethylase, is observed in metaphase II (MII) oocytes and has been proven to be crucial for maintaining genomic stability and ZGA after fertilization[70]. This indicates that *Kdm4a*-mediated H3K9me3 demethylation might occur as a stress response to artificial

activation in SCNT embryos, a possibility that needs further verification.

In our study, we performed loss-of-function studies of *Max* and *Mcrs1* by morpholino injection into zygotes. *Max* and *Mcrs1* MO injection led to an almost complete absence of blastocyst formation and aberrant differentiation of the ICM and TE. Lack of *Max* and *Mcrs1* expression resulted in large-scale epigenomic turbulence, as numerous genes related to chromatin modifications were significantly affected. However, we noted that *Max* MO embryo development was not hindered until the morula stage, while nearly half of *Mcrs1* MO embryos were arrested at each embryonic stage (Fig. 5b). These results indicate both identical and distinct regulatory mechanisms of MAX and MCRS1 during early embryonic development.

Moreover, why the candidate factors were transcriptionally repressed in SCNT embryos remains unknown. We observed no significant differences between SCNT and fertilized embryos in H3K9me3 levels at the loci of *Mcrs1, Max*, and *Mycn*. Other dysregulated histone modifications, such as H3K27me3, or DNA methylation level may be responsible for the defective expression of these factors. There might also exist other factors that directly or indirectly inhibit their transcription.

Notably, *Max* is expressed mainly in 1C and 2C embryos and is modestly reactivated in blastocysts. *Max* MO and *Max^{-/-}* embryos exhibited arrest at E3.5-E5.5, emphasizing the importance of the MAX reservoir at cleavage stages and the timing of its functional window. Despite the abundant storage of Max in 1C-E2C SCNT embryos, our *Max* OE strategy at the L2C stage demonstrated that proper reactivation of MAX in the ICM and TE is also pivotal for embryonic development, especially during the peri-implantation period. Notably, MAX can repress germ cell-related genes in mESCs and is associated with the expression of PRC1, DNMTs and SETDB1[71]. The underlying connection between MAX-mediated repression and lineage commitment during early embryonic development requires further investigation.

In contrast, the expression level of *Mcrs1* gradually increased from the E2C stage, peaked at the 4C stage, and remained high until the blastocyst stage (Supplementary Fig. 5a). This suggests that MCRS1 might be an overall regulator throughout early embryonic development. Previous research showed that MCRS1 is indispensable for epiblast formation and can induce naïve pluripotency in vitro[56,62]. In our study, *Mcrs1* OE ameliorated various defects in cumulus cell-derived SCNT embryos, improving blastocyst formation and quality, implantation capability, and even the birth rate. Both the lack of MCRS1 in fertilized embryos and *Mcrs1* OE in SCNT embryos influenced the expression or epigenetic state of various epigenetic modifiers (Fig. 5f and Supplementary Fig. 6l), indicating the improvement of SCNT embryo development by *Mcrs1* OE may be accomplished by the rescue of the whole epigenome including H3K9me3, which needs further validation. On the other hand, the improvements in postimplantation development and SCNT efficiency after *Mcrs1* overexpression are probably achieved by improved regulation of the transition between the naïve and primed pluripotent states, a possibility that requires further in-depth inquiry. Notably, the activation of *Mcrs1* is drastically blocked and delayed in cumulus cell-derived SCNT embryos, while *Mcrs1* OE at the L2C stage exactly recapitulated its expression trend, which likely resulted in a better rescue phenotype compared to that of *Max* OE embryos. Examination of H3K9me3 modification after *Mcrs1* OE in distinct somatic cell-derived SCNT embryos suggested that *Mcrs1* could consistently influence differential H3K9me3 allocation, but excess expression of *Mcrs1* may also impede the proper deposition of H3K9me3. Moreover, MCRS1 is not the only decisive TF in lineage-specific H3K9me3 deposition. Further verification of the predicted factors in this study may help reveal the mechanism of epigenetic specificity in cell fate determination.

## Methods

### Experimental model and subject details
The specific pathogen-free grade mice (SPF) grade mice, including C57BL/6j, DBA/2 and B6D2F1 mice, were housed in the animal facility at Tongji University, Shanghai, China. Mice were housed in individually ventilated cages with an environment with temperature ranging from 23 to 27°C, humidity of 30–45%, and a light cycle with 12 h of light and 12 h of darkness. All the mice had free access to food and water. All experiments were performed in accordance with the University of Health Guide for the Care and Use of Laboratory Animals and were approved by the Biological Research Ethics Committee of Tongji University (TJAB04021104).

The B6D2F1 hybrid mice (8–10 weeks old) were obtained from mating female C57BL/6j mice with male DBA/2 mice.

### Embryo collection
To obtain MII oocytes and preimplantation embryos, female B6D2F1 or C57BL/6j mice (8–10 weeks old) were superovulated by injection with 7 IU of pregnant mare serum gonadotropin (PMSG) followed by injection of 5 IU of human chorionic gonadotropin (hCG) (San-Sheng Pharmaceutical) 48 h later.

MII oocytes were retrieved from the dissected oviducts of superovulated B6D2F1 female mice at 14 h post-hCG injection for subsequent SCNT. To obtain fertilized embryos, superovulated C57BL/6j female mice were mated with male C57BL/6j mice. Then, the zygotes were collected from the oviducts of the female mice at 20 h after hCG injection and were cultured in G-1 PLUS medium (Vitrolife) until the blastocyst stage.

### Donor cell preparation
Cumulus-oocyte complexes (COCs) were collected from oviducts 14 h after hCG injection and treated with bovine testicular hyaluronidase to obtain dissociated cumulus cells and oocytes, respectively. Immature Sertoli cells were collected from the testes of 3- to 5-day-old B6D2F1 male mice. Testicular masses were incubated in PBS containing 1 mg/ml collagenase at 37°C for 10 min to obtain the dissociated Sertoli cells.

### SCNT procedure
B6D2F1 mice were used as oocyte donors, and cumulus cells or immature Sertoli cells were used as nuclear donors. MII oocytes were enucleated in Chatot-Ziomek-Bavister (CZB) medium containing 5 μg/mL cytochalasin B (CB, Sigma-Aldrich) by a piezo-driven pipette (PiezoXpert, Eppendorf) connected to an Olympus inverted microscope (IX73). The donor cells were transferred into enucleated oocytes by direct injection. The reconstructed oocytes were then cultured in CZB medium for 1 h before activation treatment. The cloned embryos were activated by 6 h of incubation in $Ca^{2+}$-free CZB containing 1 mM $SrCl_2$ and 5 μg/mL cytochalasin B. Reconstructed embryos were thoroughly washed and cultured in G-1 PLUS medium at 37°C in 5% $CO_2$.

### Knockdown of *Suv39h2* in SCNT embryos
Three siRNAs targeting *Suv39h2* were mixed in nuclease-free water to a working concentration of 20 μM per siRNA. Sequences of siRNAs were listed in Supplementary Table 1. Oocytes were injected with approximately 10 pL of the siRNA mixture using a piezo-driven micromanipulator. The injected oocytes underwent SCNT after 20 min of recovery. Embryos were collected at 6 hpa for H3K9me3 CUT&RUN.

### In vitro transcription of mRNA and mRNA injection
The WT cDNA sequences of *Usp17la, Usp17lb, Usp17lc, Usp17ld, Usp17le, Max* and *Mcrs1* were inserted into T7-driven vectors. Sequences of primers were listed in Supplementary Table 2. mRNAs were synthesized with the mMESSAGE mMACHINE T7 Ultra Kit (Thermo Fisher Scientific, AM1345) according to the manufacturer's instructions. RNA was purified by phenol-chloroform extraction and ethyl alcohol

precipitation. The integrity of the synthesized mRNA was confirmed by electrophoresis. The precipitated RNA was suspended in $H_2O$ at several concentrations and stored at −80 °C until use. For mRNA injection, enucleated oocytes or L2C cloned embryos were injected with approximately 10 pL of mRNA using a piezo-driven micromanipulator. In some experiments, the mRNA was diluted to an ideal concentration before injection.

## Morpholino targeting of *Max*, *Mcrs1*, and *Mycn* in fertilized embryos

Isolated zygotes were injected with either 1 mM morpholino oligonucleotides targeting *Max, Mcrs1* and *Mycn* or a 25-base random control morpholino. The targeting sites of the morpholinos were listed in Supplementary Table 3. All morpholinos were designed and synthesized with Gene Tools (https://www.gene-tools.com/). Injected embryos were cultured in G-1 PLUS medium at 37 °C under 5% $CO_2$. The knockdown (KD) efficiency was determined at the morula or blastocyst stage. ICM and TE from zygotes injected with 0.3 mM *Max* MO or 0.2 mM *Mcrs1* MO were collected for Smart-seq2.

## Reverse transcription and quantitative RT-PCR analysis

For quantitative RT-PCR analysis of the morpholino KD efficiency, cDNA from 10 blastomeres was synthesized and amplified using Smart-seq2. Quantitative RT-PCR was performed using SYBR Premix Ex Taq, and signals were detected with the ABI7500 Real-Time PCR System. *H2afz* was used as an endogenous control.

## Inhibition of DNA replication by aphidicolin

To block DNA synthesis, SCNT embryos were treated with 3 µg/mL aphidicolin upon activation for 6 h. Samples were then collected for H3K9me3 immunofluorescence staining.

## Immunofluorescence staining

Embryos were fixed with 4% paraformaldehyde for 1 h at room temperature and then permeabilized with 0.2% Triton X-100 for 1 h at room temperature. The samples were incubated with primary antibodies against H3K9me3 (Active Motif, 39161), Max (Proteintech, 10426-1-AP), Mcrs1 (Sigma, HPA039057), Oct4 (Santa Cruz, sc-5279) or Cdx2 (Abcam, ab76541) for 2 h at room temperature. After washing three times with 1% bovine serum albumin (BSA, Sigma-Aldrich) in PBS, the samples were incubated with the appropriate secondary antibodies for 1 h at room temperature. Nuclei were stained with 4′,6-diamidino-2-phenylindole (DAPI, Sigma-Aldrich). All stained samples were observed using a Zeiss LSM880 confocal microscope. Images were processed and quantified in ImageJ software.

## Embryo transfer

The 2-cell-stage control or mRNA-injected SCNT embryos were transferred into the oviducts of pseudopregnant female mice. For implantation ability assessment, a cesarean section was carried out on E10.5, and the embryos were stripped from the decidua. For the cloning efficiency assay, a cesarean section was carried out on E19.5; the pups were removed from the uterus, and fluid was quickly removed from their air passages. The pups were photographed, weighed, and monitored to see if they could breathe autonomously. The placentas of the E19.5 pups were also weighed and photographed.

## Sample harvest for ChIP-seq, Smart-seq2

Cumulus cells and embryos at each of the following stages were collected: 6 hpa and 14 hpa; late 2-cell, 4-cell, and 8-cell stages; morula stage; and the E4 blastocyst stage (ICM and TE). For the embryos collected before the blastocyst, the zona pellucida was removed with 0.5% pronase E (Sigma-Aldrich, P8811), and the embryos were then incubated in $Ca^{2+}$-free CZB medium for 5 min. Polar bodies were removed by gentle pipetting using a fire-polished glass needle. For ICM

and TE isolation, to eliminate tight cell-cell junctions, the zona pellucidae removed blastocyst were incubated in $Ca^{2+}$-free CZB medium for 20 min. ICM (12–15 µm) and TE (18–20 µm) were then separated by micromanipulation using needles with an inner diameter of 20 µm, according to their distinct morphology and spatial position. Spatially, since TE comprises the outer layer of a blastocyst while ICM lies inside, the cells separated at first were TE, with a much flabbier cell-cell junction than that of ICM. Morphologically, TE possesses more smooth cell membrane surface and mostly larger cell size. To ensure the accuracy of our manual separation to the greatest extent, any cell with an indistinct phenotype that might misguide our judgment was discarded.

## ULI-NChIP library construction and sequencing

For ULI-NChIP-seq, 300–500 blastomeres or cumulus cells were used per reaction, and two replicates were performed for each stage of SCNT embryos. All isolated blastomeres were washed three times in PBS solution containing 0.5% BSA to prevent contamination. The ULI-NChIP procedure was performed as previously described[10,36]. One microgram of an anti-H3K9me3 antibody was used for each immuno-precipitation reaction.

The sequencing libraries were constructed using the KAPA HyperPrep Kit (KAPA Biosystems, KK8504) for the Illumina platform (kk8504) following the manufacturer's instructions. Paired-end 150 bp sequencing was performed on the NovaSeq (Illumina) platform at Berry Genomics Co., Ltd.

## CUT&RUN library construction and sequencing

CUT&RUN was conducted following a modified published protocol. Fifty oocytes or embryos were used per reaction, and one or two replicates were performed for each stage. The samples were resuspended in 600 µL of room temperature wash buffer (HEPES pH = 7.5, 20 mM; NaCl, 150 mM; spermidine, 0.5 mM; BSA, 0.1%; and Roche complete protease inhibitor) by gentle pipetting. Five microliters of concanavalin-coated magnetic beads were transferred to a microfuge tube containing a 3× volume of cold binding buffer (HEPES-KOH pH = 7.9, 20 mM; KCl, 10 mM; $CaCl_2$, 1 mM; $MnCl_2$, 1 mM). The beads were washed twice in 1 mL of cold binding buffer and resuspended in 300 µL of binding buffer. The beads were added to the cells by gentle vortexing and incubated for 10 min (min) at room temperature. Bead-bound nuclei were blocked with 1 mL of cold blocking buffer (HEPES pH = 7.5, 20 mM; NaCl, 150 mM; spermidine 0.5 mM; BSA, 0.1%; EDTA pH = 8.0, 2 mM; PIC and digitonin, 0.1%) and incubated for 5 min at room temperature. The beads were washed in 1 mL of cold Dig-Wash Buffer (Digitonin, 0.01%) and resuspended in 250 µL of Dig-Wash Buffer. One microgram of the anti-H3K9me3 antibody was added by gentle vortexing of the bead-bound cells in 250 µL of cold Dig-Wash Buffer. The samples were incubated with rotation at 4 °C for 2 h or overnight. The samples were washed twice in 1 mL of cold Dig-Wash Buffer and resuspended in 250 µL of cold Dig-Wash Buffer with a 1:500 dilution of pA-MNase (a gift from the Steven Henikoff lab). The samples were incubated with rotation at 4 °C for 1 h and washed twice in 1 mL of cold Dig-Wash Buffer. The supernatant was discarded, and the samples were resuspended in 150 µL of cold Dig-Wash Buffer. The samples were equilibrated to 0 °C on ice for 5 min, and 3 µL of $CaCl_2$ (100 mM) was added to initiate cleavage. The reactions were stopped by the addition of 150 µL of STOP Buffer (NaCl, 200 mM; EDTA pH = 8.0, 20 mM; EGTA pH = 8.0, 4 mM; digitonin, 0.02%; RNase A, 50 µg/mL; glycogen, 20 µg/mL) after 15 min of digestion. The samples were incubated at 37 °C for 20 min to digest RNA and release DNA fragments. The samples were centrifuged at 16,000 × g for 5 min, after which the supernatants were transferred to a new microfuge tube, while the pellets and beads were discarded. Three microliters of 10% SDS and 2.5 µL of proteinase K (20 mg/mL) were added to wash the samples and incubated at 70 °C for 10 min. DNA was purified by phenol-chloroform

extraction followed by ethanol purification. The sequencing libraries were constructed using the KAPA Hyper Prep Kit for the Illumina platform (kk8504) following the manufacturer's instructions. Paired-end 150-bp sequencing was performed on the NovaSeq (Illumina) platform at Berry Genomics and Novogene.

## Smart-seq2 library construction and sequencing

For Smart-seq2, 10 blastomeres were used per reaction, and three replicates were performed for each stage. All isolated blastomeres were washed three times in PBS solution containing 0.5% BSA to prevent contamination. RNA-seq libraries were constructed using the Covaris DNA shearing protocol for Smart-seq sequence library construction as previously described[72]. Briefly, RNAs with a poly-adenylated tail were captured, reverse transcribed and pre-amplified. After fragmentation, the sequencing libraries were constructed using the KAPA HyperPrep Kit for the Illumina platform following the manufacturer's instructions. Paired-end 150-bp sequencing was performed on the NovaSeq (Illumina) platform at Berry Genomics and Novogene.

## ChIP-seq data processing

The H3K9me3 ULI-NChIP-seq data for oocyte samples, sperm samples, and PN3, PN5, 2C, 4C, 8C, morula, ICM and TE samples from fertilized embryos were obtained from a previous publication (GSE97778)[26], and the H3K9me3 ULI-NChIP-seq data for CC samples and 6 hpa, 14 hpa, 2C, 4C, 8C, morula, ICM and TE samples from SCNT embryos were generated in this study. The H3K9me3 CUT&RUN data for 6 hpa wild-type embryos and siRNAs targeting *Suv39h2* in SCNT embryos were generated in this study.

All ULI-NChIP-seq and input data reads were aligned to the mm10 reference genome using BWA[73] with the mem command after trimming with Cutadapt[74] with the parameters "--trim-n -q 25,25 -m 75 -a AGATCGGAAGAGC -A AGATCGGAAGAGC". Signal tracks for each sample were generated using the deepTools[75] bamCoverage function with normalization to RPKM (Reads Per Kilobase Million) and removal of PCR duplicates. The results (bigWig files) are shown in the track view generated with Integrative Genomics Viewer (IGV)[76] (Fig. 1b). To check the reproducibility of the ChIP-seq experiments, we calculated the Pearson correlation coefficients of the H3K9me3 signals between biological replicates in the merged peaks.

CUT&RUN data were processed similarly to ULI-NChIP-seq data, with additional alignment to the sacCer3 (*Saccharomyces cerevisiae*) reference genome as the spike-in standard. Signal tracks were generated using the deepTools bamCoverage function with normalization to 1 million reads based on the sequencing depth of the spike-in standard.

## Identification of H3K9me3 peaks

As the H3K9me3 peak coverage is affected by the sequencing depth, the alignment files were first merged using SAMtools[77], and PCR duplicates were then removed and downsampled to the same number of reads (60 million) using the MarkDuplicates and DownsampleSam commands in Picard[78] tools (http://broadinstitute.github.io/picard/). The H3K9me3 peaks were called with MACS2[79] with the parameter "--broad" and over the input data. To estimate the difference in H3K9me3 across stages in SCNT embryos and fertilized embryos, the H3K9me3 peaks were split into 1-kb windows, and then the H3K9me3 signal in these windows was calculated as the log2-transformed H3K9me3/Input ratio (pseudocount = 1).

## Annotation and definition

The enrichment of H3K9me3 peaks in annotated genomic regions was calculated using the Homer[80] annotatePeaks command. The heatmap and profile line plot for the H3K9me3 signal around annotated genomic regions (i.e., CGIs, YY1, and H3K9me3 peaks) were generated using deepTools.

The genomic regions (i.e., LTRs, CGIs, promoters) with H3K9me3 modification were defined as those with the intersection of the H3K9me3 peak at the midpoint using the BEDtools[81] intersect command with the parameters "-r -e -f 0.5". The promoters were defined as the regions encompassing 1 kb upstream and downstream of the TSS.

## Analysis of reprogrammed and unreprogrammed H3K9me3

The reprogrammed H3K9me3 regions before ZGA were identified as the windows of H3K9me3 peaks that were significantly present in CCs and were not present at 6 hpa or 14 hpa; conversely, the unreprogrammed H3K9me3 regions were identified as the H3K9me3 peak windows that were significantly present in CCs and had not been removed at 14 hpa or the 2C stage (Supplementary Fig. 3b). These H3K9me3 regions were used to find enriched motifs by using the findMotifsGenome.pl script in the Homer tool, and the motifs with the greatest difference in enrichment significance between the reprogrammed H3K9me3 regions and the unreprogrammed H3K9me3 regions are shown (Fig. 3a).

## RNA-seq data processing

We generated RNA-seq data from samples including CCs, 6 hpa, 14 hpa, E2C, L2C, 4C, 8C, morula, ICM, TE, oeMax-ICM, oeMax-TE, oeMcrs1-ICM and oeMcrs1-TE from SCNT embryos and MO-Max-ICM, MO-Max-TE, MO-Mcrs1-ICM and MO-Mcrs1-TE from fertilized embryos. The RNA-seq data for oocyte samples and PN5, E2C, L2C, 4C, 8C and ICM samples from fertilized embryos were obtained from a previous publication[82] (GSE71434). All RNA-seq data were mapped to the mm10 reference genome after trimming with Cutadapt using HISAT2[83] with the parameters '--data-cufflinks --no-discordant --no-mixed --no-unal'. The TPM (transcripts per million) values of genes were obtained by StringTie[84]. Differential gene expression analysis was performed using DESeq2[85].

## Identification of minor ZGA, major ZGA and MGA genes

To define the ZGA genes more accurately, we used strict filters to identify the minor ZGA, major ZGA and MGA genes by DESeq2. First, genes with a normalized expression level of greater than 50 normalized counts in oocytes were removed as potential maternally expressed genes, and then genes whose expression was greater than 100 normalized counts at the PN5 or E2C stage with greater than 2-fold upregulation of and a q-value of less than 0.01 compared to that in oocytes were identified as minor ZGA genes. Next, among the remaining genes, the genes upregulated in L2C embryos under the same conditions as above were identified as major ZGA genes. Similarly, the genes upregulated in 4 C embryos were identified as MGA genes. Finally, we identified 299 minor ZGA genes, 1249 major ZGA genes and 499 MGA genes.

The genes in each of the above three classes of genes were classified as fully, partially and restrictedly activated genes in SCNT embryos. Restrictedly activated genes were identified as those with a fold change (fertilized/SCNT) > 4, fully activated genes were identified as those with a fold change (fertilized/SCNT) < 2, and the remaining genes were termed partially activated genes.

## Repeat analysis

H3K9me3 signal and expression at repeats were quantified using the Homer tool. In brief, mapped data were processed with the make-TagDirectory command in Homer with the parameter "-keepOne". The tag files of samples by biological replicate were analyzed using the "analyzeRepeats.pl repeats" command in the Homer tool with RPKM normalization. For the analysis of repeat families, the "-condenseL2" parameter was used to combine the read counts of the repeats with the same family annotation.

To quantify the DNA methylation level in MERVL-int and MT2_Mm, we first calculated DNA methylation levels (from a previous publication[10], GSE108711) on each copy of the annotated repeats,

summed the values for the same annotated repeats, and then averaged the level with respect to the copy number in the genome.

## Gene ontology analysis

Functional enrichment analysis of gene sets was performed using the ClusterProfiler[86] tool in R. The 5 functions with the smallest q-values in each cluster of genes were selected and compared in all samples. The q-values were not greater than 0.05 (Fig. 5f and Supplementary Fig. 2 g, 6l).

## Clustering analysis of H3K9me3 domains in the first lineage differentiation

To determine the differences in H3K9me3 between SCNT embryos and fertilized embryos in the first cell lineage differentiation, we first merged the domains of the H3K9me3 peaks in SCNT-ICM, SCNT-TE, fertilized-ICM and fertilized-TE and then recalculated the H3K9me3 signals in these regions by biological replicate. This signal matrix was used for principal component analysis (PC1 and PC2 are shown in Fig. 4a and Supplementary Fig. 4a) in R. The first two columns of the eigenvalues of the rotation matrix determined by the analysis were used to filter out the domains with the most positive and negative loading (eigenvector value > 3, $n = 3427$; Supplementary Fig. 4b), and the signal matrices of these filtered domains were displayed as a heatmap and classified into 7 clusters using K-means clustering (Fig. 4b).

Based on the 7 clusters of H3K9me3 regions, the Euclidean distance for a specific region "i" between SCNT and fertilized embryos based on their H3K9me3 signals in the ICM and TE was calculated, denoted as $A_i$. The distance between Mcrs1 OE SCNT embryos and fertilized embryos was denoted as $B_i$, and the distance between Mcrs1 OE SCNT embryos and SCNT embryos was denoted as $C_i$. If $A_i > B_i$, it indicates that Mcrs1 OE has a repairing effect on the H3K9me3 of the i region, accounting for approximately 73% of the total regions. Among them, regions with $\log2(B_i/C_i) < 1$ were defined as successfully rescued regions, accounting for approximately 33% of the total regions.

## Analysis of TF-binding sites associated with H3K9me3

To investigate the transcription factors associated with the abnormal lineage differentiation of H3K9me3 in SCNT embryos, we collected publicly available binding peak data of TFs, histone enzymes and chromatin remodelers in mouse ESCs from the GEO database, as previously described[26]. The numbers of binding sites of these factors overlapping with the H3K9me3-specific regions in SCNT-ICM, SCNT-TE, fertilized-ICM, and fertilized-TE were calculated, the aberrant H3K9me3 in SCNT embryos was evaluated by determining fold change values, and significance was calculated using Fisher's exact test with the total number of the H3K9me3-specific regions as the background value. To confirm the differences in H3K9me3 at the candidate TF-binding sites, we recalculated the H3K9me3 signals at these sites and assessed the significance of the differences using the Kruskal-Wallis test followed by Dunn's multiple comparison test.

## Statistics and reproducibility

Error bars in the graphical data represent the Standard Error of the Mean (SEM). For all the presented boxplots, the center line represents the median value, the lower and upper hinges correspond to the first and third quartiles (the 25th and 75th percentiles), the whisker extends from the hinge to the largest or smallest value at most 1.5 * IQR (interquartile range) of the hinge. The one-tailed Wilcoxon rank-sum test was used to analyze the significant difference between different groups in Fig. 3b. The two-tailed Wilcoxon rank-sum test was used to analyze the significant difference between different groups in Figs. 2c, 6h and Supplementary Fig. 4f, 6n. The two-tailed Student's t test was used to analyze the significant difference between different groups in Figs. 2d, 3f, 3g, 3h, 5b, 6c, 6e, 6f and Supplementary Fig. 2d, 5b–e, 6b,

6d–g, 6i. Two-tailed Fisher's exact test was used to analyze the data in Fig. 4d and Supplementary Fig. 4e, 6m, 6o. ChIP-seq and RNA-seq experiments were performed two to six times for each group. The precise numbers of replicates and the data qualities were summarized in a Source Data file.

## Reporting summary

Further information on research design is available in the Nature Portfolio Reporting Summary linked to this article.

## Data availability

The H3K9me3 ULI-NChIP-seq data for oocytes, sperm, PN3, PN5, 2C, 4C, 8C, morula, ICM and TE of fertilized embryos were obtained from a previous publication[26] (GSE97778), and the H3K9me3 ULI-NChIP-seq data for CC, 6 hpa, 14 hpa, 2C, 4C, 8C, morula, ICM and TE of SCNT embryos were generated in this study. The RNA-seq data for oocytes, PN5, E2C, L2C, 4C, 8C and ICM of fertilized embryos were obtained from a previous publication[82] (GSE71434). The RNA-seq data for SCNT embryos and other embryos were generated in this study. The DNA methylation data for SCNT and fertilized embryos were obtained from a previous publication[10] (GSE108711). The ChIP-seq data of Mcrs1 in mESCs were obtained from a previous publication[87] (GSE51746). The raw sequence data reported in this paper have been deposited into the Gene Expression Omnibus (GEO). The accession number for the SCNT H3K9me3 ChIP-seq, CUT&RUN, and RNA-seq data generated in this paper is GSE195762. Source data are provided with this paper.

## Code availability

Custom codes used for the analysis reported in this study are available at [https://github.com/rysterzhu/H3K9me3-in-SCNT].

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

## Acknowledgements

This work was primarily funded by the National Natural Science Foundation of China (Grant No. 31820103009 to S.G.), the National Key R&D Program of China (Grant Nos. 2022YFA1106000 to X.L., 2021YFC2700300 to S.G., 2020YFA0112500 to C.L., 2022YFC2702200 to S.G. and C.L., 2021YFA1102900 to X.L. and 2021YFA1100300 to C.J.), and the National Natural Science Foundation of China (Grant Nos. 32070802 to X.L., 31721003 to S.G., 32100645 to M.C., 32270858 to C.J. and 32100652 to G.Y.). This work was also supported by the Shanghai Rising-Star Program (20QA409700 to X.L.), the China Postdoctoral Science Foundation (Grant Nos. 2022M722424 to Q.Z., 2020M681383 to M.C., BX2021218 to M.C., 2022M722420 to S.S., 2021M692438 to G.Y., 2022T150482 to G.Y.), the Science and Technology Commission of Shanghai Municipality (Grant Nos. 19JC1415300 and 21JC1405500 to S.G.), the Fundamental Research Funds for the Central Universities (Grant No. 22120230292 to S.G., X.L. and C.J.) and Peak Disciplines (Type IV) of Institutions of Higher Learning in Shanghai to S.G.

## Author contributions

X.L. and S.G. conceived and designed the experiments. R.X. and C.L. conducted the experiments with assistance from M.C., Yuyan Z., L.Y., Z.S., X.Z., Q.S., Y.L., X.K., Yanhong Z. and H.W. Q.Z. and C.J. designed and analyzed the data with assistance from S.S. and G.Y. R.X., Q.Z., X.L., C.L. and S.G. wrote the manuscript.

## Competing interests

The authors declare no competing interests.
