## [Peer Review File · Nature Communications]

Unreprogrammed H3K9me3 prevents minor zygotic genome activation and lineage commitment in SCNT embryosREVIEWER COMMENTS

Reviewer #1 (Remarks to the Author):

Xu and co-authors performed genome-wide profiling of H3K9me3 in cumulus-derived SCNT embryos and identified how the H3K9me3 marks were inherited from the donor cells in a stage-specific manner. Importantly, the H3K9me3 marks most affected the transcription of minor ZGA genes, which lead to developmental arrest at the 2-cell stage, compromising subsequent activation of repetitive elements, especially LTRs. These are important findings to characterize the epigenomic feature of SCNT embryos, but given the data from numerous mouse SCNT studies to date, including the author groups', these results are to be expected. The most important finding in this study may be the indistinct H3K9me3 deposition pattern between ICM and TE in SCNT blastocysts, which was caused by downregulation of specific transcription factors including Max and Mcrs1. The role of Mcrs1 was particularly important; overexpression of Mcrs1 in SCNT embryos normalized the distinct H3K9me3 patterns of ICM and TE in blastocysts, and indeed significantly improved clonal efficiency. Overall, cloning experiments and genome-wide H3K9me3 analysis are accurate, and conclusions were correctly drawn from the results obtained. However, since the Mcrs1 results are particularly impactful in the SCNT as well as developmental biology areas, it will be important to confirm whether the same results can be obtained using other somatic donors, such as immature Sertoli cells or fibroblasts. It is not necessary to reproduce all the results with other somatic donor type(s), but specifically, the indistinct H3K9me3 deposition pattern of ICM and TE and the effect of Mcrs1 overexpression on it should be confirmed. Another question that arises from the results of this paper is why Max and Mcrs1 are downregulated in SCNT embryos. According to the authors view, this was not related to H3K9me3. It needs to be examined whether this can be explained in terms of SCNT biology. If possible, it should be shown in the experimental results, and if difficult, it should at least be mentioned in the discussion.

Minor points:

1. Line 63: For a paper describing the barrier of H3K9me3 to cell fate change, we recommend citing a more recent paper, Hada et al. *Genes Dev* 2020 (PMID: 34992147)
2. Line 118 (Figure 1F): "0%" should be included on the Y axis in some way.
3. Line 147: Why was MII oocytes analyzed for comparison? This is an epigenetic status before activation and SCNT embryos do not contain the MII chromosomes.
4. Line 243: ICM or TE, which is more responsible for the indistinct H3K9me deposition between them in SCNT blastocysts, when compared with the pattern in fertilized embryos?
5. Line 265: Why were Max, Mcrs1 and Mycn repressed in SCNT embryos? By H3K9me3?
6. Line 303: Max OE result would better be presented as a supplemental data.
7. Fig. 2C: It is better to mention that the data was from fertilized embryos, because the title of Figure 2 means that the contained figures show SCNT data.
8. Fig. S3 (legend): Although mentioned in the Methods section, it would better mention in the legend that the data were from GSE108711.

Reviewer #2 (Remarks to the Author):

In SCNT, H3K9me3 in donor cells has been widely recognized as a barrier to SCNT-mediated reprogramming since the discovery of highly improved developmental rate of

SCNT embryos upon Kdm4d overexpression (Matoba et al. 2014). However, the success rate is still limited. The authors previously reported the dynamics of H3K9me3 in normal early mouse embryos (Wang et al. 2018), and in a continuation of that work, Xu et al. profiled H3K9me3 distribution in SCNT embryos. H3K9me3 was aberrantly enriched at minor ZGA genes and affected their expression, as expected from the papers published before (there are several papers showing that Dux and its target minor ZGA genes are downregulated in SCNT embryos in a H3K9me3-dependent manner). The authors then nicely showed that H3K9me3 distribution in ICM and TE are severely compromised in SCNT blastocysts. The lack of H3K9me3 deposition at the genomic regions containing Mcrs1- and Max-binding motifs as well as their downregulated expression prompted the authors to test the effect of Mcrs1 or Max OE in SCNT embryos. Mcrs1 OE in SCNT embryos significantly improved the development of SCNT embryos, suggesting that Mcrs1 might be involved in reprogramming H3K9me3 for proper lineage allocation.

I see that the provided datasets for SCNT embryos would be widely useful, with unexpected and interesting findings on SCNT blastocysts. However, I feel that the results presented are limited only to support the authors' conclusions. I believe that a more unbiased and impartial data analysis would strengthen the message of this paper and therefore suggest the following points to improve the manuscript.

Major points

1. Is there any plausible explanation/discussion why only a small fraction of minor ZGA genes were affected in their expression in SCNT embryos although H3K9me3 was generally deposited on these genes? Also, it will be better to mention that the affected genes at minor ZGA are the Dux-target genes.
2. It is totally unclear how Mcrs1 functions in H3K9me3 reprogramming. The authors show that H3K9me3 deposition was compromised at Mcrs1-binding sites in SCNT blastocysts. However, the effect of Mcrs1 OE on the remodeling of H3K9me3 at the Mcrs1-binding sites is not shown. It is at least necessary to investigate if Mcrs1 OE preferentially alters H3K9me3 levels at its target sites as Mcrs1 OE tended to decrease rather than increase H3K9me3 levels (Fig. 6I).
3. Fig. 6G-I - Effects of Mcrs1 OE on lineage-specific H3K9me3 should be shown as in Fig. 4B, and it will be important to quantitatively show how much H3K9me3 levels are corrected.
4. Related to the comments above, it is highly possible that Mcrs1 OE exerts beneficial effects on SCNT embryo development indirectly affecting H3K9me3 enrichment and that the observed corrected distribution of H3K9me3 can be a secondary effect. In this context, there might be a possibility that Mcrs1 directly induce or repress key cell fate determination genes. Further data analysis and discussion considering these possibilities would be necessary.
5. Fig. 3B – The effect of Kdm4b OE was limited to minor ZGA gene loci although it induces the global loss of H3K9me3 levels. To understand the effect of Kdm4b OE on the development of SCNT embryos, an analysis for genome-wide H3K9me3 levels, including the intergenic regions, repetitive elements and all gene loci, would be necessary. Along the same lines, the authors can investigate the effect of Kdm4b OE at the reprogramming-resistant YY1-binding sites (Fig. S3E).

Minor points

1. Fig. 1F – Please include a plot based on the comparison of H3K9me3 distribution between fertilized embryos and donor cells (i.e., data for fertilized embryo H3K9me3 overlapped with

CC). With this additional plot, it will be much easier to see the donor-derived defective distribution of H3K9me3 in SCNT embryos.

2. Fig. 2D – Please include the donor cell H3K9me3 data. This will enable to see how much de novo deposition of H3K9me3 was suppressed by Suv39h2 KD.

3. Fig. S3 - H3K9me3-reprogrammed or unreprogrammed regions at 6 and 14 hpa and the 2-cell stage should be shown by heatmaps as in Fig. S3E. Also, as the statement “We then classified H3K9me3 regions based on whether they could be reprogrammed until 14 hpa” seems to be inconsistent with the data shown. Please rephrase it or describe more clearly.

4. Please indicate bin/gene numbers in each figure when applicable.

5. Fig. 4A,B - Inclusion of 8-cell and morula H3K9me3 data from fertilized and SCNT embryos will be necessary to see the defects in H3K9me3 reprogramming in SCNT blastocysts (although Fig. S1E shows high similarity between SCNT morula and ICM/TE).

6. Fig. 4C and S4B - What is the difference between plots highlighted in red, blue and gray?

7. Fig. 5E,F - It is uncertain how ICM and TE cells were isolated from Mcrs1 MO embryos although no blastocysts were formed (Fig. 5B).

8. Immunofluorescence on Mcrs1 and Max during early development would be informative in respect to their roles on lineage allocation.

9. Line 265 - Some description of the current knowledge on the function of Max, Mcrs1 and Mycn would be helpful as it is unclear why the authors focused on them in terms of epigenetic reprogramming.

Reviewer #3 (Remarks to the Author):

H3K9me3-marked domains present a barrier to reprogramming to pluripotency for both mouse and human somatic cell nuclear transfer (SCNT). It is known that overexpression of H3K9me3 can dramatically improve the blastocyst formation rate of SCNT. In this paper, the authors showed how H3K9me3 is programmed during the early development of SCNT embryos. By comparing with fertilized embryos, widespread failure of H3K9me3 reprogramming is observed at early stages, resulting in defective activation of minor ZGA genes and repeats. The finding is consistent with the previous studies. Furthermore, abundant de novo H3K9me3 is found at CpG islands (CGIs) and promoters. Moreover, they identified Mcrs1 as a regulator to help configure lineage-specific H3K9me3 in ICM and TE, and overexpression of Mcrs1 can improve the development rate of SCNT embryos.

The study made a number of interesting observations and involved a substantial amount of work on SCNT embryos. The H3K9me3 data here, together that the datasets generated in fertilized embryos from the same lab, also provide an important resource for understanding the role of H3K9me3 in epigenetic reprogramming in embryogenesis. It helps elucidate the mechanisms underlying previous observations on how KDM facilitates SCNT reprogramming and confirmed the previous model that a key role of KDM is to remove somatic K9me3 and to facilitate the activation of minor ZGA genes. One caveat of the current paper is that at times conclusions in the paper were not well supported, or data were not closely connected

to the conclusions. Some of the data and analyses were difficult to understand. The identification of *Mcrs1* as a facilitator for SCNT development is interesting, although the evidence of how H3K9me3 reprogramming is related to *Mcrs1* is still unclear. Please see detailed comments below.

Major comment:

1. The abstract needs to be better and more accurately phrased. It is unclear what the authors meant by H3K9me3 being “persistently redundant” throughout SCNT development. From my reading from the manuscript, K9me3 undergoes dynamic reprogramming in SCNT embryos. It would be misleading to say that “with most marks inherited from donor cells”. This also contradicts the authors’ statement that “H3K9me3 marks that overlapped with CCs largely disappeared between 1-8C stages (page 5).
2. The overlap with CC can be used as a reference to infer the inheritance of H3K9me3 but cannot be overstated, given K9me3 in any two cell types would have a certain degree of overlap due to the preferential enrichment of H3K9me3 at repeats. Such “background” overlap should be considered when estimating the level of inheritance from CC.
3. The similar H3K9me3 patterning between ICM and TE in SCNT embryos is striking. Can the authors provide validation to rule out cross-contamination?
4. Figure 1E, the loss and gain of H3K9me3 in SCNT development at each stage seem substantial. This seems to contradict to the statement that H3K9me3 is relatively stable after ZGA based on this figure.
5. The author showed *Suv39h2* knockdown could reduce de novo established H3K9me3 at CGIs in Figures 2E and F. However, the reduction of H3K9me3 is rather weak and is unclear if this is reproducible. Moreover, does *Suv39h2* knockdown impact SCNT development?
6. Figure 6I, the authors only showed a few snapshots for *Mcrs1* OE results. It is unclear to what extent can *Mcrs1* overexpression restore H3K9me3 patterning in SCNT embryos. Fig. 6G seems to support this idea. However, I suggest the authors to map the K9me3 data to figure 4B which would more clearly demonstrate this result.
7. Fig. 4C. Why is K9me3 fertilized embryo/SCNT predominantly greater than 0 (x-axis), while we can clearly see both up and down of H3K9me3 in Fig. 4B?
8. Fig. 4B. Could H3K9me3 in SCNT reflect H3K9me3 state at a preceding stage due to delayed development, which may explain why it is so different from that in fertilized embryos?
9. Can the authors comment/speculate on the function of de novo H3K9me3 at CGIs?
10. It is unclear if the K9me3 effects by *Mcrs1* is direct or indirect, especially given *Mcrs1* and Max MO injection can affect *Suv39h1*. This should be commented.
11. Figure 4B, it is interesting that a large proportion of H3K9me3 established in fertilized embryos was not properly established in SCNT embryos either. These regions should also be discussed.

12. Page 8, Line 219, the delayed activation of repeats only correlates with DNA methylation, not “caused”, at least based on data in this paper. Such overstatements are not limited to this example.

13. Fig. 3I. Why K9me3 increased upon Kdm4b OE at L2C for MERVL? Please clarify.

14. Fig. 3J. Why many MERVLs showed more K9me3 in fertilized embryos compared to SCNT embryos (X-axis, >0)? In addition, it seems the upregulated and downregulated MERVLs are comparable.

Minor comment:

1. Supplemental figures in general are too small to read.

2. Figure 1C, 14hpa showed more H3K9me3 increase than 6hpa. What are the regions with increased H3K9me3?

3. Figure 1E, more H3K9me3 disappeared at 2C than that at later stages in SCNT embryos. Could the author examine what are these regions?

4. SCNT embryos are a mixture of embryos capable to successfully develop to blastocyst and that not capable to do so. Although it is difficult to conduct H3K9me3 at single blastomere levels, it is necessary to discuss how this would affect the paper's conclusions in Discussion.

5. Fig. 1B, the K9me3 enrichment is quite low at Zscan4 and MERVL in both CC and fertilized embryos in 1-2C embryos, contrary to the authors' statements. This should be clarified.

6. Figure 1D appeared earlier than Figure 1C.

7. Does KDM4D overexpression rescue MERVL expression defects too in SCNT embryos?

8. Figure 2C, the aphidicolin-treated sample showed only one pronucleus?

9. Line 358. A citation is missing.

10. Lines 370-375. It is unclear what the authors meant by “artificial activation” when referring to the Kdm4a's role in MII oocytes.

**Point to point response:**

We appreciate the reviewers' time and effort in reviewing our manuscript, "Unreprogrammed
H3K9me3 prevents minor ZGA and lineage commitment in SCNT embryos". Their constructive
feedback would enable us to substantially improve our work. We performed additional
experiments and computational analyses based on the suggestions of reviewers. The major
changes were summarized as follows.

• We performed additional analyses to further dissect how Mcrs1 benefits lineage-specific
H3K9me3 deposition in SCNT blastocysts.

• We examined the protein expression trend of Max and Mcrs1 during fertilized and SCNT
embryos development and found decreased Max and Mcrs1 signals in SCNT morulae and
blastocysts.

• We validated the reproducibility of *siSuv39h2* to block the *de novo* H3K9me3 deposition
in 6hpa SCNT embryos using CUT&RUN.

• We added analysis about the function of *Kdm4b* OE on H3K9me3 deposition and
retrotransposon expression.

• We improved textual quality of the revised manuscript by describing more carefully
about the data, adding additional methodological details, and providing more thoughtful
discussion to show the advance of the data for the research filed.

We have listed the point-to-point response to the comments raised by all three reviewers. We hope
that with this sufficient revision, our study would be suitable to be published in Nature
Communications. The reviewer's comments are in blue followed by our response in black.

**Reviewers' Comments to the Authors:**

**Reviewer #1:**

Xu and co-authors performed genome-wide profiling of H3K9me3 in cumulus-derived SCNT
embryos and identified how the H3K9me3 marks were inherited from the donor cells in a stage-
specific manner. Importantly, the H3K9me3 marks most affected the transcription of minor ZGA
genes, which lead to developmental arrest at the 2-cell stage, compromising subsequent activation
of repetitive elements, especially LTRs. These are important findings to characterize the
epigenomic feature of SCNT embryos, but given the data from numerous mouse SCNT studies to
date, including the author groups', these results are to be expected. The most important finding
in this study may be the indistinct H3K9me3 deposition pattern between ICM and TE in SCNT
blastocysts, which was caused by downregulation of specific transcription factors including Max
and Mcrs1. The role of Mcrs1 was particularly important; overexpression of Mcrs1 in SCNT

embryos normalized the distinct H3K9me3 patterns of ICM and TE in blastocysts, and indeed
significantly improved clonal efficiency.

Major points:

1. Overall, cloning experiments and genome-wide H3K9me3 analysis are accurate, and conclusions
were correctly drawn from the results obtained. However, since the *Mcrs1* results are particularly
impactful in the SCNT as well as developmental biology areas, it will be important to confirm
whether the same results can be obtained using other somatic donors, such as immature Sertoli
cells or fibroblasts. It is not necessary to reproduce all the results with other somatic donor type(s),
but specifically, the indistinct H3K9me3 deposition pattern of ICM and TE and the effect of *Mcrs1*
overexpression on it should be confirmed.

We thank the reviewer for seeing the value of our H3K9me3 datasets in cumulus-derived SCNT
embryos. To confirm whether the *Mcrs1* results can be obtained using other somatic donors, we
performed the SCNT procedure using immature Sertoli cells and examined H3K9me3 in Sertoli-
derived SCNT ICM and TE (Figure R1a). As expected, compared with fertilized embryos, Sertoli-
derived SCNT blastocysts also exhibited an inappreciable distinction between ICM and TE on
H3K9me3 establishment (Figure R1a). Consistently, we then examined and found poor enrichment
of lineage-specific H3K9me3 at TF binding sites in Sertoli-derived SCNT ICM and TE (Figure R1b).
However, we observed quite a few TFs whose binding sites were highly enriched with abnormal
H3K9me3 signal in Sertoli-derived SCNT ICM/TE, which is contrary to cumulus-derived SCNT (Figure
R2c, d). Meanwhile, *Mcrs1* did not rank high in the TF candidates list. These results suggest an
alternative regulatory mechanism underlying Sertoli cell-mediated reprogramming of H3K9me3,
which might be independent of *Mcrs1*.

For further verification, we examined the expression level of *Mcrs1* during Sertoli-derived SCNT
embryo development. The expression of *Mcrs1* initiates in 2-cell fertilized embryos, whose
activation is seriously repressed and delayed to the 8-cell stage in cumulus-derived SCNT embryos
(Figure S5A). By contrast, our data showed that *Mcrs1* is already sufficient expressed in 2-cell and
4-cell Sertoli-derived SCNT embryos (Figure R1e). Moreover, H3K9me3 in *Mcrs1* overexpressed (OE)
Sertoli-derived SCNT blastocysts persisted close to the control ones (Figure R1e). Notably, *Mcrs1*
OE dramatically strengthened the lineage differentiation of H3K9me3 deposition on various TF
binding sites, yet leading the Sertoli-derived SCNT blastocysts to a direction opposite that of the
fertilized ones (Figure R1f). Furthermore, *Mcrs1* OE have no obvious effect on Sertoli-derived
cloning efficiency (figure R1h). These results indicate that *Mcrs1* is highly related to lineage-specific
H3K9me3 establishment, but excess *Mcrs1* may also impede the proper deposition of H3K9me3
and the regulation of *Mcrs1* varies in different somatic cell-mediated reprogramming.

**Figure R1. Distinct regulation of *Mcrs1* in Sertoli-derived SCNT reprogramming.**

a. PCA of H3K9me3 in ICM and TE of fertilized, cumulus-derived SCNT and Sertoli-derived SCNT embryos.

b-d. Volcano plots showing the H3K9me3 signal on TF binding sites in ICM and TE of fertilized and Sertoli-
derived SCNT embryos.

c. Bar plot showing the relative expression level of *Mcrs1* during Sertoli-derived SCNT embryo development.
RT-qPCR analysis was performed in fertilized and Sertoli-derived SCNT embryos. The relative expression
levels of *Mcrs1* relative to *H2afz* were compared between two groups at the same stage. Data are presented
as the means \pm SEM. ns, no significance. * $p < 0.05$, ** $p < 0.01$, and **** $p < 0.0001$ by Student's t-test for
comparison.

f. Unrooted dendrogram of H3K9me3 signals in ICM and TE of fertilized and Sertoli-derived SCNT embryos
(control and *Mcrs1* OE groups).

84 g. Volcano plots showing the H3K9me3 signal on TF binding sites in ICM and TE of fertilized and Sertoli-
85 derived SCNT embryos (control and *Mcrs1* OE groups).

86 h. Bar plot showing the birth rate of full-term Sertoli-derived SCNT pups after *Mcrs1* OE. N indicates the
87 number of transferred 2-cell embryos. R indicates the number of replicates. Data are presented as the means
\pm SEM. ns, no significance.

2. Another question that arises from the results of this paper is why *Max* and *Mcrs1* are
downregulated in SCNT embryos. According to the authors view, this was not related to H3K9me3.
It needs to be examined whether this can be explained in terms of SCNT biology. If possible, it
should be shown in the experimental results, and if difficult, it should at least be mentioned in the
discussion.

After examining the H3K9me3 levels on the *Mcrs1*, *Max*, and *Mycn* loci in both SCNT and fertilized
embryos, we observed no significant differences. However, analysis of other histone modifications
and DNA methylation levels revealed that abnormal H3K27me3 in SCNT embryos (as evident in
data from another unpublished project in our group, as shown in Figure R2) may be responsible
for the defective expression of these genes. Nevertheless, other factors that directly or indirectly
inhibit their expression cannot be entirely ruled out. We have added the statement in the
discussion part of the revised manuscript.

**Figure R2. Unreprogrammed H3K27me3 may repress the activation of *Max*, *Mcrs1*, and *Mycn*.**

Genome Browser view of H3K27me3 signals around the *Mcrs1*, *Max*, and *Mycn* loci in the fertilized (green)
and SCNT (red) embryos. The signals are presented as the RPKM of the H3K27me3 signals.

**Minor points:**

1. Line 63: For a paper describing the barrier of H3K9me3 to cell fate change, we recommend citing
a more recent paper, Hada et al. *Genes Dev* 2020 (PMID: 34992147)

We appreciate the reviewer for the suggestion. The paper has been cited in the revised manuscript.

2. Line 118 (Figure 1F): "0%" should be included on the Y axis in some way.

We thank the reviewer's suggestion. We have updated the Y-axis in the revised Fig. 1F. Moreover,
to better understand the correlation of the H3K9me3 dynamics between CC, SCNT, and fertilized
embryos, we have added the line showing the overlap of H3K9me3 between fertilized embryos
and CC in this figure.

3. Line 147: Why was MII oocytes analyzed for comparison? This is an epigenetic status before
activation and SCNT embryos do not contain the MII chromosomes.

In this section, we propose the hypothesis that SCNT embryos are affected by residual Suv39h2 in
the oocyte cytoplasm and build up *de novo* H3K9me3, as shown not only by CGI (Figure S2F) but
also by the function of genes similar to oocyte (Figure S2G), which lead to the simulation of the
chromatin state in fertilized embryos and that *de novo* H3K9me3 on the promoter regions of these
genes is useful for suppressing gene function in late developmental stages. We have revised the
statement in the revised manuscript.

4. Line 243: ICM or TE, which is more responsible for the indistinct H3K9me deposition between
them in SCNT blastocysts, when compared with the pattern in fertilized embryos?

To answer the question, we investigated H3K9me3 in fertilized and SCNT at the morula stage
(revised Figure 4C) and found numerous differentially marked H3K9me3 regions between them (*eg.*
Cluster2, 4, and 7). Also, we observed a rather high similarity between SCNT morula, ICM, and TE.
These suggest that the abnormal H3K9me3 deposition in SCNT ICM/TE may already exist at the
morula stage. Besides, we noticed that fertilized TE tended to acquire a more significant amount
of lineage-specific H3K9me3 than ICM (figure S4C). However, in SCNT blastocysts, these H3K9me3
are either not established in TE or removed in ICM. Thus, TE-specific H3K9me3 deficiency might
contribute more to the indistinct H3K9me3 deposition in SCNT blastocysts.

5. Line 265: Why were *Max*, *Mcrs1* and *Mycn* repressed in SCNT embryos? By H3K9me3?

As discussed in the 2nd major point, we examined the epigenetic status at the loci of *Max*, *Mcrs1*,
and *Mycn* (Figure R2), and found that they were all covered by higher H3K27me3 signals in SCNT
embryos, which may repress their expression.

6. Line 303: *Max* OE result would better be presented as a supplemental data.

We thank the reviewer for the suggestion. We have added the statistical result of the
developmental efficiency of *Max* OE SCNT embryos in the revised Figure S6E.

7. Fig. 2C: It is better to mention that the data was from fertilized embryos, because the title of
Figure 2 means that the contained figures show SCNT data.

We appreciate the reviewer's thoughtful suggestion. We have modified the title of Figure 2C and
also clarified that the data was from fertilized embryos in the figure legend of the revised
manuscript.

8. Fig. S3 (legend): Although mentioned in the Methods section, it would better mention in the
legend that the data were from GSE108711.

We appreciate the reviewer's advice and have added the statement in the figure legend of the
revised manuscript as suggested.

**Reviewer #2:**

In SCNT, H3K9me3 in donor cells has been widely recognized as a barrier to SCNT-mediated
reprogramming since the discovery of highly improved developmental rate of SCNT embryos upon
Kdm4d overexpression (Matoba et al. 2014). However, the success rate is still limited. The authors
previously reported the dynamics of H3K9me3 in normal early mouse embryos (Wang et al. 2018),
and in a continuation of that work, Xu et al. profiled H3K9me3 distribution in SCNT embryos.
H3K9me3 was aberrantly enriched at minor ZGA genes and affected their expression, as expected
from the papers published before (there are several papers showing that Dux and its target minor
ZGA genes are downregulated in SCNT embryos in a H3K9me3-dependent manner). The authors
then nicely showed that H3K9me3 distribution in ICM and TE are severely compromised in SCNT
blastocysts. The lack of H3K9me3 deposition at the genomic regions containing Mcrs1- and Max-
binding motifs as well as their downregulated expression prompted the authors to test the effect
of Mcrs1 or Max OE in SCNT embryos. Mcrs1 OE in SCNT embryos significantly improved the
development of SCNT embryos, suggesting that Mcrs1 might be involved in reprogramming
H3K9me3 for proper lineage allocation.

I see that the provided datasets for SCNT embryos would be widely useful, with unexpected and
interesting findings on SCNT blastocysts. However, I feel that the results presented are limited only
to support the authors' conclusions. I believe that a more unbiased and impartial data analysis
would strengthen the message of this paper and therefore suggest the following points to improve
the manuscript.

**Major points**

1. Is there any plausible explanation/discussion why only a small fraction of minor ZGA genes were
affected in their expression in SCNT embryos although H3K9me3 was generally deposited on these
genes? Also, it will be better to mention that the affected genes at minor ZGA are the Dux-target
genes.

We thank the reviewer for pointing this out. Indeed, only 10% of minor ZGA genes were completely
repressed in SCNT embryos (revised Figure S3G). Our data showed that H3K9me3 in CC is highly
correlated with the repression of minor ZGA genes in SCNT embryos. However, the majority of
H3K9me3-marked regions in CCs showed reduced H3K9me3 signal and were identified as
reprogrammable H3K9me3 regions (revised Figure S3B), which suggested that these H3K9me3
modifications may not be strong enough to directly and completely block the transcription of the
marked ZGA genes. In addition, a certain number of SCNT embryos can develop towards blastocyst
despite with defect of ZGA, suggesting the presence of multiple compensation mechanisms during
early development. Moreover, we strongly agree with the reviewer that the affected minor ZGA
genes are Dux-target genes and that the H3K9me3 deposition on these minor ZGA genes is also
non-negligible. We have added these statements in the revised manuscript.

2. It is totally unclear how Mcrs1 functions in H3K9me3 reprogramming. The authors show that

H3K9me3 deposition was compromised at *Mcrs1*-binding sites in SCNT blastocysts. However, the
 effect of *Mcrs1* OE on the remodeling of H3K9me3 at the *Mcrs1*-binding sites is not shown. It is at
 least necessary to investigate if *Mcrs1* OE preferentially alters H3K9me3 levels at its target sites as
 *Mcrs1* OE tended to decrease rather than increase H3K9me3 levels (Fig. 6I).

We appreciate the reviewer's suggestion. We investigated H3K9me3 levels at *Mcrs1* binding sites
 in ICM and TE after *Mcrs1* OE. We found that *Mcrs1* OE led to strengthened H3K9me3 enrichment
 on *Mcrs1* targets in both lineages and partially recovered the differential H3K9me3 allocation in
 SCNT blastocysts (Figure R3). These results have been updated in the revised Figures 6H and 6I.

a

b

**Figure R3. H3K9me3 levels on binding sites of *Mcrs1* in ICM and TE after *Mcrs1* OE.**

a. Box plot showing H3K9me3 levels on *Mcrs1* binding sites in ICM and TE of SCNT, fertilized, and *Mcrs1* OE
 SCNT blastocysts.

b. Genome Browser view of H3K27me3 signals around the *Mcrs1* binding sites in the fertilized (green), SCNT
 (red) and *Mcrs1* OE SCNT (orange) embryos. The signals are presented as the RPKM of the H3K27me3 signals.

3. Fig. 6G-I - Effects of *Mcrs1* OE on lineage-specific H3K9me3 should be shown as in Fig. 4B, and
 it will be important to quantitatively show how much H3K9me3 levels are corrected.

We appreciated the reviewer's suggestion. To confirm this, we showed the data of *Mcrs1* OE ICM
 and TE as in Figure 4B. We found that about 33% of H3K9me3 regions were successfully corrected
 by *Mcrs1* OE (Figure R4a). We focused on H3K9me3-marked protein-coding genes to further

investigate the biological function of *Mcrs1*. Importantly, *Mcrs1* positively regulated H3K9me3 on
 multiple genes encoding histone modifiers, such as *Trim28*, *Dnmt3a*, *Kmt2c*, and *Ezh2* (Figure R4c,
 upper panel). These indicate that in addition to directly inducing H3K9me3 deposition (Figure R3),
 *Mcrs1* may also regulate lineage-specific H3K9me3 in an indirect manner, by affecting genes
 related to lineage commitment. However, the lineage-specific H3K9me3 deposition in promoters
 of some developmental genes which are critical for lineage commitment can't be rescued by *Mcrs1*
 OE (eg. *Gata3*, *Gata6*, *Fgfr1*) (Figure R4b, lower panel), which indicate there are other critical
 factors need to be validated. We have updated the discussion in the revised manuscript and the
 analysis in the revised Figures S6I and S6J.

**Figure R4. *Mcrs1* OE rescued H3K9me3 levels on binding sites of *Mcrs1* in ICM and TE after *Mcrs1* OE.**

a. Heat map showing the scaled H3K9me3 signals in ICM and TE of fertilized and SCNT (control and *Mcrs1*
 OE) embryos. The regions were classified the same as in Figure 4B.

b. Go-term analysis of genes marked by *Mcrs1* OE-rescued and un-rescued H3K9me3.

4. Related to the comments above, it is highly possible that *Mcrs1* OE exerts beneficial effects on
 SCNT embryo development indirectly affecting H3K9me3 enrichment and that the observed
 corrected distribution of H3K9me3 can be a secondary effect. In this context, there might be a
 possibility that *Mcrs1* directly induce or repress key cell fate determination genes. Further data
 analysis and discussion considering these possibilities would be necessary.

We agree with the reviewer that *Mcrs1* may affect the enrichment of H3K9me3 both directly and
 indirectly (Figures R3 and R4). On the other hand, we analyzed the functions of the genes affected
 by knocking down *Mcrs1* and *Max* in fertilized embryos (Figure 5F). The lack of *Mcrs1*
 downregulated numerous genes related to various epigenetic modifications in addition to
 H3K9me3, indicating the global regulatory functions of *Mcrs1* on the epigenome. We have updated

the analysis and discussion in the revised manuscript.

5. Fig. 3B – The effect of *Kdm4b* OE was limited to minor ZGA gene loci although it induces the
global loss of H3K9me3 levels. To understand the effect of *Kdm4b* OE on the development of SCNT
embryos, an analysis for genome-wide H3K9me3 levels, including the intergenic regions, repetitive
elements and all gene loci, would be necessary. Along the same lines, the authors can investigate
the effect of *Kdm4b* OE at the reprogramming-resistant YY1-binding sites (Fig. S3E).

We appreciate the reviewer's suggestion. We investigated the H3K9me3 levels in 14hpa and L2C
SCNT embryos after *Kdm4b* OE and found that *Kdm4b* globally removed SCNT-specific H3K9me3,
without bias of certain genomic regions (Figure R5a, b). As the reviewer suggested, we examine
the H3K9me3 levels around the binding sites of Yy1 and found that the unrpH3K9me3 regions
were successfully removed after *Kdm4b* OE (Figure R5c). We have updated the figures in the
revised manuscript.

**Figure R5. *Kdm4b* OE eliminated genome-wide excessive H3K9me3 in SCNT embryos.**

a. Heatmap showing the H3K9me3 signal in SCNT embryos (control and *Kdm4b* OE) before 2C, only regions
 overlapped with the H3K9me3 peaks in CC were presented. The regions were clustered by the H3K9me3
 signal using the k-means function into 5 groups.

b. Bar plot showing the enrichment of H3K9me3 peaks in LINE, LTR and promoter regions. Each cluster
 corresponds to the one in a.

c. Heatmap showing the relative H3K9me3 signal around the Yy1 binding sites.

**Minor points**

1. Fig.1F – Please include a plot based on the comparison of H3K9me3 distribution between

fertilized embryos and donor cells (i.e., data for fertilized embryo H3K9me3 overlapped with CC).
With this additional plot, it will be much easier to see the donor-derived defective distribution of
H3K9me3 in SCNT embryos.

We appreciate the reviewer's suggestion and have added the data showing the overlap of
H3K9me3 between fertilized embryos and CC in the revised Figure 1F.

2. Fig. 2D – Please include the donor cell H3K9me3 data. This will enable to see how much *de novo*
deposition of H3K9me3 was suppressed by *Suv39h2* KD.

We would like to clarify that the control and *siSuv39h2* data were generated by CUT&RUN, while
the others were generated via ULI-NChIP-seq. We separately presented the H3K9me3 signals in CC
and SCNT embryos generated by different methods (Figure R6). Notably, we observed a
comparable *de novo* H3K9me3 level in 6hpa SCNT embryos by NChIP-seq (Figure R6a) and
CUT&RUN (Figure R6b). Based on this, we could confirm that *de novo* H3K9me3 was appreciably
suppressed by *Suv39h2* KD. However, the signals of inherited and erased H3K9me3 seemed to vary
from the sequencing methods. Thus, we thought it inappropriate to put them in the same figure.
For verification of the influence of *Suv39h2* KD on *de novo* H3K9me3, we have additionally
repeated the KD experiments twice and the results were consistent. We have updated the result
in the revised Figure 2D.

**Figure R6. *Suv39h2* KD considerably reduced *de novo* H3K9me3 in 6hpa SCNT embryos.**

a. Box plot showing the H3K9me3 signals in CC and 6hpa SCNT embryos at *de novo*, inherited and erased
H3K9me3 regions. The data were generated by NChIP-seq. ****p < 0.0001 by the Wilcoxon rank-sum
test.

b. Box plot showing the H3K9me3 signals in control and *siSuv39h2* SCNT embryos at *de novo*, inherited
and erased H3K9me3 regions. ****p < 0.0001 by the Wilcoxon rank-sum test. ns, no significance.

3. Fig. S3 - H3K9me3-reprogrammed or unreprogrammed regions at 6 and 14 hpa and the 2-cell
stage should be shown by heatmaps as in Fig. S3E. Also, as the statement "We then classified
H3K9me3 regions based on whether they could be reprogrammed until 14 hpa" seems to be

inconsistent with the data shown. Please rephrase it or describe more clearly.

We have shown the heatmap of H3K9me3-reprogrammed and unreprogrammed regions in the
revised Figure S3B. To avoid misunderstanding, we have rephrased the statement in the revised
manuscript.

4. Please indicate bin/gene numbers in each figure when applicable.

We have submitted all available data for bar, point, and heatmap plots as source data for each
figure.

5. Fig. 4A,B - Inclusion of 8-cell and morula H3K9me3 data from fertilized and SCNT embryos will
be necessary to see the defects in H3K9me3 reprogramming in SCNT blastocysts (although Fig. S1E
shows high similarity between SCNT morula and ICM/TE).

We thank the reviewer's suggestion. We have included the H3K9me3 data of fertilized and SCNT
morulae and found numerous differentially marked H3K9me3 regions between them (*eg.* Cluster
2, 4, and 7) (revised Figure 4C). The result supported that the defects in H3K9me3 reprogramming
may already exist at the morula stage, as H3K9me3 in SCNT morula showed a similar pattern with
SCNT ICM/TE.

6. Fig. 4C and S4B - What is the difference between plots highlighted in red, blue and gray?

We are sorry for the omission. The red plots represent the potential TFs that may be related to
defective differential H3K9me3 deposition in SCNT blastocysts. The blue plots represent well-
known chromatin architecture-related and H3K9me3-related factors. The gray plots represent
other TFs. We have added the statement in the revised legend.

7. Fig. 5E,F - It is uncertain how ICM and TE cells were isolated from *Mcrs1* MO embryos although
no blastocysts were formed (Fig. 5B).

We thank the reviewer for pointing this out. We first tried the recommended 1 mM MO but
couldn't get any blastocyst. For RNA-seq sample collection, we performed a concentration
gradient test of both Max and *Mcrs1* Morpholinos. We finally selected 0.3 mM Max MO and 0.2
mM *Mcrs1* MO as the appropriate condition, since they resemble fertilized embryos until the
morula stage and meanwhile keep the blastocyst rate to about 30% (Figure R7). We have added
the statement in the legend and method section of the revised manuscript.

**Figure R7. Test of *Max* and *Mcrcs1* MO concentration for sample collection.**

a, b. Line charts showing the developmental rate of fertilized embryos injected with different *Max* (left) and
 *Mcrcs1* (right) MO concentrations.

c. Representative images of *Max* and *Mcrcs1* MO-injected embryos at embryonic day 3.5 (E3.5). Scale bar, 100
 332 μm .

8. Immunofluorescence on *Mcrcs1* and *Max* during early development would be informative in
 respect to their roles on lineage allocation.

We appreciate the reviewer for the suggestion. We examined the expression pattern of *Max*
 (Proteintech, 10426-1-AP) and *Mcrcs1* (Sigma, HPA039057) in fertilized and SCNT embryos (Figure
 R8). According to our immunostaining results, although both *Max* and *Mcrcs1* were already
 accumulated during early cleavages, they showed decreased protein expression levels in SCNT
 embryos at both morula and blastocyst stages. This further suggested that there might be other
 factors correlated with *Max*/*Mcrcs1* that were also dysregulated during SCNT lineage segregation.
 These results were updated in the revised Figure S5B.

**Figure R8. Immunofluorescence of Max and Mcrs1 in fertilized and SCNT early embryos.**

a. Quantification of Max and Mcrs1 immunofluorescence signal in fertilized and SCNT embryos. Data are
 presented as the means \pm SEM. Each dot represents one nuclear. * $p < 0.05$ and **** $p < 0.0001$ by Student's
 t-test for comparison.

b. Representative images of immunofluorescence of Max and Mcrs1 in fertilized and SCNT morulae and
 blastocysts. BF, bright field. DAPI stands for DNA in blue and Max/Mcrs1 is in green.

9. Line 265 - Some description of the current knowledge on the function of Max, Mcrs1 and Mycn
 would be helpful as it is unclear why the authors focused on them in terms of epigenetic
 reprogramming.

We thank the reviewer's suggestion. We have updated a brief review of previous studies on these
 TFs in the revised manuscript.

**Reviewer #3:**

H3K9me3-marked domains present a barrier to reprogramming to pluripotency for both mouse
and human somatic cell nuclear transfer (SCNT). It is known that overexpression of H3K9me3 can
dramatically improve the blastocyst formation rate of SCNT. In this paper, the authors showed
how H3K9me3 is programmed during the early development of SCNT embryos. By comparing with
fertilized embryos, widespread failure of H3K9me3 reprogramming is observed at early stages,
resulting in defective activation of minor ZGA genes and repeats. The finding is consistent with the
previous studies. Furthermore, abundant de novo H3K9me3 is found at CpG islands (CGIs) and
promoters. Moreover, they identified Mcrs1 as a regulator to help configure lineage-specific
H3K9me3 in ICM and TE, and overexpression of Mcrs1 can improve the development rate of SCNT
embryos.

The study made a number of interesting observations and involved a substantial amount of work
on SCNT embryos. The H3K9me3 data here, together that the datasets generated in fertilized
embryos from the same lab, also provide an important resource for understanding the role of
H3K9me3 in epigenetic reprogramming in embryogenesis. It helps elucidate the mechanisms
underlying previous observations on how KDM facilitates SCNT reprogramming and confirmed the
previous model that a key role of KDM is to remove somatic K9me3 and to facilitate the activation
of minor ZGA genes. One caveat of the current paper is that at times conclusions in the paper were
not well supported, or data were not closely connected to the conclusions. Some of the data and
analyses were difficult to understand. The identification of Mcrs1 as a facilitator for SCNT
development is interesting, although the evidence of how H3K9me3 reprogramming is related to
Mcrs1 is still unclear. Please see detailed comments below.

**Major comment:**

1. The abstract needs to be better and more accurately phrased. It is unclear what the authors
meant by H3K9me3 being “persistently redundant” throughout SCNT development. From my
reading from the manuscript, K9me3 undergoes dynamic reprogramming in SCNT embryos. It
would be misleading to say that “with most marks inherited from donor cells”. This also
contradicts the authors’ statement that “H3K9me3 marks that overlapped with CCs largely
disappeared between 1-8C stages (page 5).

We are sorry for the confusion. We do agree with the reviewer that H3K9me3 underwent dynamic
reprogramming in SCNT embryos, according to Figures 1E and 1F. Meanwhile, we noticed that
H3K9me3 covered more regions in SCNT embryos than in fertilized ones at each corresponding
stage (except 2C) (Figure 1C) and SCNT-specific H3K9me3 regions always dominated (Figure 1D).
This led us to believe that H3K9me3 is highly dynamic as well as redundant during SCNT embryo
development. To avoid misleading, we change the ‘persistently redundant’ to ‘excess’ in the
revised manuscript and replaced the ‘with most marks inherited from donor cells’ with ‘with most
marks showed SCNT-specific deposition’

2. The overlap with CC can be used as a reference to infer the inheritance of H3K9me3 but cannot
be overstated, given K9me3 in any two cell types would have a certain degree of overlap due to
the preferential enrichment of H3K9me3 at repeats. Such “background” overlap should be
considered when estimating the level of inheritance from CC.

We thank the reviewer for pointing this out. To consider the “background” overlap between the
two cell types, we updated Figure 1F with the addition of fertilized data. H3K9me3 peaks also show
nearly 50% overlap between CC and fertilized embryos after the 2-cell stage, which is similar to
SCNT embryos. On the other hand, the overlap level with CC is extremely high before the 2-cell
stage in SCNT embryos, which can reflect the H3K9me3 inherited from CC was gradually removed.
We have changed the statement in the revised manuscript.

3. The similar H3K9me3 patterning between ICM and TE in SCNT embryos is striking. Can the
authors provide validation to rule out cross-contamination?

Our protocol for ICM and TE separation is fixed in all previous subjects, including profiling of histone
modifications in mouse and human pre-implantation embryos (Liu et al., 2016; Wang et al., 2018;
Xu et al., 2022). Also, the same separation protocol was also used in fertilized ICM/TE separation,
which shows a distinct H3K9me3 pattern. We listed and compared the expression levels of typical
lineage-specific markers in ICM and TE of both fertilized and SCNT. The result showed that the
lineage-biased expression trend is consistent in SCNT embryos, which further supported that the
potential cross-contamination may have little impact on our analysis (Figure R9). We have provided
a detailed description in the method section of our manuscript at “Sample Harvest for CHIP-seq,
Smart-seq2”: To eliminate tight cell-cell junctions, the zona pellucidae removed blastocyst were
incubated in Ca²⁺-free CZB medium for 20 minutes. ICM (12-15 μm) and TE (18-20 μm) were then
separated by micromanipulation using needles with an inner diameter of 20 μm, according to their
distinct morphology and spatial position. Spatially, since TE comprises the outer layer of a
blastocyst while ICM lies inside, the cells separated at first were basically TE, with a much flabbier
cell-cell junction than that of ICM. Morphologically, TE possesses more smooth cell membrane
surface and mostly larger cell size. To ensure the accuracy of our manual separation to the greatest
extent, any cell with an indistinct phenotype that might misguide our judgment was discarded.

**Figure R9. Consistent expression trend of lineage-specific markers in ICM and TE of fertilized and SCNT**
 **embryos.**

Box plot showing the expression levels of lineage-specific markers in ICM and TE of fertilized and SCNT
 embryos. * $p < 0.05$, ** $p < 0.01$, *** $p < 0.001$ and **** $p < 0.0001$ by Student's t-test for comparison. ns, no
 significance.

4. Figure 1E, the loss and gain of H3K9me3 in SCNT development at each stage seem substantial.
 This seems to contradict to the statement that H3K9me3 is relatively stable after ZGA based on
 this figure.

We are sorry for the misleading, here we want to say the total amount of H3K9me3 is relatively
 stable after ZGA, for the amount of established and disappeared H3K9me3 is much the same based
 on Figure 1E. The identical fraction of established and disappeared H3K9me3 domains during
 embryonic development after the 2-cell stage is also consistent with our observation in the
 fertilized embryos (Wang et al., 2018). To avoid misunderstanding, we have removed the
 statement about 'H3K9me3 is relatively stable after ZGA' in the revised manuscript.

5. The author showed Suv39h2 knockdown could reduce de novo established H3K9me3 at CGIs in
 Figures 2E and F. However, the reduction of H3K9me3 is rather weak and is unclear if this is
 reproducible. Moreover, does Suv39h2 knockdown impact SCNT development?

To validate the reproducibility of *siSuv39h2*, we performed two batches of knockdown
 experiments and assessed the H3K9me3 levels in Control and *siSuv39h2* 6hpa SCNT embryos using
 CUT&RUN. The results were consistent with our previous findings (Figure R10). We merged the
 data of two replicates in the revised manuscript. Besides, as observed previously in fertilized
 embryos (Burton et al., 2020), *siSuv39h2* did not affect SCNT development.

**Figure R10. Knockdown of *Suv39h2* led to a reduction of the H3K9me3 signal at CGIs.**
a. Box plot showing the H3K9me3 signal (log2 transformed RPKM) in *de novo*, inherited and erased
H3K9me3 regions in *siSuv39h2* and control SCNT embryos at 6 hpa. P value by the Wilcoxon rank-sum
test.
b. Distributions of H3K9me3 signal around CpG-islands in *siSuv39h2* and control SCNT embryos at 6 hpa.
c. Genome Browser view of H3K9me3 around the *Wwp2* gene locus in *siSuv39h2* and control SCNT
embryos at 6 hpa.
459 d. Box plot showing the blastocyst rate of SCNT E4 embryos after knocking down *Suv39h2*. The data are
460 represented as the means \pm SEMs. ns, no significance.
e. Representative images of E4 SCNT embryos after knocking down *Suv39h2*. Scale bar, 100 μ m.

6. Figure 6I, the authors only showed a few snapshots for *Mcrs1* OE results. It is unclear to what
extent can *Mcrs1* overexpression restore H3K9me3 patterning in SCNT embryos. Fig. 6G seems to
support this idea. However, I suggest the authors to map the K9me3 data to figure 4B which would
more clearly demonstrate this result.

We appreciate the reviewer's suggestion. We have mapped the *Mcrs1* OE data to Figure 4B and
represented the heatmap (Figure R3). We also investigated the change of H3K9me3 levels on
*Mcrs1* binding sites after *Mcrs1* OE (Figure R4). We also updated the analysis in the revised
manuscript. Please also review our discussion in the 2nd and 3rd major points of Reviewer#2.

7. Fig. 4C. Why is K9me3 fertilized embryo/SCNT predominantly greater than 0 (x-axis), while we
can clearly see both up and down of H3K9me3 in Fig. 4B?

We thank the reviewer's question. We would like to clarify that not all the clusters of H3K9me3 in
Figure 4B exhibit enrichment of binding sites of transcription factors, which usually bind regulatory
regions, such as promoters. We analyzed the genomic distribution of the 7 clusters of H3K9me3
shown in Figure 4B (Figure S4B). SCNT embryos showed the deficient establishment of H3K9me3
regions in clusters 4 and 6, which are enriched in promoter regions in fertilized embryos. On the
contrary, SCNT embryos possessed higher H3K9me3 signals in cluster 5, which are enriched in LINE
and LTR regions. These are consistent with the results in figure 4C (revised figure 4D) which showed
higher enrichment of H3K9me3 on TF binding site in fertilized embryos.

8. Fig. 4B. Could H3K9me3 in SCNT reflect H3K9me3 state at a preceding stage due to delayed
development, which may explain why it is so different from that in fertilized embryos?

We thank the reviewer's suggestion. We can't exclude the hypothesis that some H3K9me3 defect
is due to the delayed development, but the reprogramming of SCNT embryos is largely different
from fertilized embryos, for the SCNT embryos need to remove the somatic epigenetic marks
during development. According to the unrooted cluster analysis in Figure S1E, the H3K9me3
pattern is largely different between SCNT and fertilized embryos, which can't be explained only by
the delayed development. Also, we have added morula H3K9me3 data from fertilized and SCNT
embryos in the revised Figure 4C. We observed a dramatically high similarity of H3K9me3 between

SCNT morula and ICM/TE samples. Notably, Cluster 2, 4, 6, and 7 are regions where SCNT morulae
failed to establish/remove H3K9me3, and the deficiency was maintained in blastocysts. The data
suggested that the defects in H3K9me3 reprogramming already exist at the morula stage.

9. Can the authors comment/speculate on the function of de novo H3K9me3 at CGIs?

Previous research addressed the biological significance of *de novo* H3K9me3 catalyzed by Suv39h2
in fertilized embryos (Burton et al., 2020). *De novo* H3K9me3 appeared immediately after
fertilization on the paternal nuclei, and paternally enriched H3K9me3 regions displayed open
chromatin configuration during early cleavages. Depletion of *Suv39h2* led to an increased
proportion of accessible promoters at the 8-cell stage, indicating that *de novo* H3K9me3 at
paternal genome primes genomic regions for chromatin compaction at later developmental stages.

10. It is unclear if the K9me3 effects by Mcrs1 is direct or indirect, especially given Mcrs1 and Max
MO injection can affect Suv39h1. This should be commented.

We have performed further analysis to dissect how *Mcrs1* affects H3K9me3 allocation. *Mcrs1* OE
in SCNT embryos led to strengthened H3K9me3 signals on *Mcrs1* targeting regions in both lineages
which may show the direct function. *Mcrs1* also influenced the expression or epigenetic state of
various epigenetic modifiers, including H3K9me3 modifiers which indicates the indirect function
of *Mcrs1*. We have added these analyses and discussions to the revised manuscript. Please also
review our results and discussion in the 2nd and 3rd major points of Reviewer#2 (Figures R3, 4).

11. Figure 4B, it is interesting that a large proportion of H3K9me3 established in fertilized embryos
was not properly established in SCNT embryos either. These regions should also be discussed.

As discussed in the 7th point of the reviewer's major comments and the 3rd major point of
reviewer#2, we analyzed the genomic distribution of the 7 clusters of H3K9me3 shown in Figure
4B (Figure S4B). H3K9me3 from clusters 4 and 6 were unestablished in SCNT embryos. These
H3K9me3 regions were specifically enriched in promoters and marked genes related to
developmental maturation and embryonic organ development (Figures S6K, L). This indicates that
these dysregulated H3K9me3 regions may be correlated with abnormal gene regulation during
lineage commitment.

12. Page 8, Line 219, the delayed activation of repeats only correlates with DNA methylation, not
"caused", at least based on data in this paper. Such overstatements are not limited to this example.

We thank the reviewer for pointing this out. The statement has been altered as suggested in the
revised manuscript.

13. Fig. 3I. Why K9me3 increased upon Kdm4b OE at L2C for MERVL? Please clarify.

In Figure 3H, the H3K9me3 signal of MERVL increased at the L2C and 4C stages of fertilized
 embryos. Based on Figure 3G, we speculate that the activation and suppression (H3K9me3
 increased) of MERVL in SCNT embryos is delayed compared to fertilized embryos, which can be
 partially rescued by *Kdm4b* OE. We further analyzed the effect of *Kdm4b* OE on H3K9me3-related
 methyltransferases. The result showed that the H3K9me3 signal was decreased on the *Setdb1*
 locus at 14hpa (Figure R11a). Correspondingly, the expression of *Setdb1* was partially restored at
 the 2C stage (Figure R11b). This might be one of the direct reasons why the H3K9me3 signal on
 MERVL increased at the L2C stage after *Kdm4b* OE. However, we acknowledge that further
 research is needed to further understand the underlying mechanisms.

**Figure R11. *Kdm4b* OE is correlated to an increased expression level of *Setdb1*.**

a. Genome Browser view showing H3K9me3 signal around *Setdb1* locus in *Kdm4b* OE and control SCNT 14hpa
 embryos.

b. Bar plot showing the expression of *Setdb1* in fertilized, SCNT and *Kdm4b* OE embryos.

14. Fig. 3J. Why many MERVLs showed more K9me3 in fertilized embryos compared to SCNT
 embryos (X-axis, >0)? In addition, it seems the upregulated and downregulated MERVLs are
 comparable.

We are sorry that the X-axis and Y-axis were wrongly stated. The right ones are updated in the
 revised manuscript with “log2(H3K9me3 of SCNT/Fertilized)” and “log2(expression of SCNT
 L2C/Fertilized L2C)”, respectively. Generally, the number of H3K9me3-marked MERVLs is much
 more abundant in SCNT embryos especially at the L1C stage (X-axis > 0).

**Minor comment:**

1. Supplemental figures in general are too small to read.

The font size in the supplemental figure has been zoomed in as suggested.

2 & 3. Figure 1C, 14hpa showed more H3K9me3 increase than 6hpa. What are the regions with
increased H3K9me3? Figure 1E, more H3K9me3 disappeared at 2C than that at later stages in SCNT
embryos. Could the author examine what are these regions?

Figure S1C shows that the H3K9me3 peaks at 14hpa remained enriched in LTR and LINE regions,
without an increase in other genomic regions. We further examined the average length and
number of H3K9me3 peaks from CC to 2-cell SCNT embryos (Figure R12). The results indicate that
the increased H3K9me3 peaks at 14hpa were mainly due to the lengthening of peaks, rather than
an enlarged number of peaks. At the 2-cell stage, the H3K9me3 peaks were removed and
shortened (Figure R12). We propose that this is the overlapping result of the processes of
H3K9me3 removal from the donor nuclei and the establishment of H3K9me3 imitating the
fertilized embryo. In Figure 2, we demonstrated that after SCNT activation, Suv39h2 affected the
*de novo* establishment of H3K9me3, and this process was likely to continue until 14 hpa. In addition,
it can be seen from the fertilized embryo that H3K9me3 establishment before the 2-cell stage is
continuous (Figure 1C), while H3K9me3 from CC is constantly removed. Another possibility is that
the cell cycle stage of 6 hpa and 14 hpa cells is different, but since we do not have direct
experimental verification, this part of the hypothesis is not included in this paper.

**Figure R12. The dynamic number and length of H3K9me3 peaks in CC and SCNT embryos.**

a. The changes of the number (blue box) and mean length (orange line) of H3K9me3 peaks in CC, 6hpa, 14hpa
and 2C SCNT embryos.

b. Genome Browser view showing the length of H3K9me3 peaks in CC, 6hpa, 14hpa and 2C SCNT embryos.

4. SCNT embryos are a mixture of embryos capable to successfully develop to blastocyst and that
not capable to do so. Although it is difficult to conduct H3K9me3 at single blastomere levels, it is
necessary to discuss how this would affect the paper's conclusions in Discussion.

We strongly agree with the reviewer that the examination of H3K9me3 in SCNT embryos is still a
mixed result regardless of the developmental capacity. Single-cell-based ChIP-seq technology will
help us focus only on the samples showing unreprogrammed epigenomes. Moreover, given that
the deficient lineage-specific H3K9me3 deposition may already appear in SCNT morulae (or maybe
even earlier), the single-blastomere investigation can greatly empower us to confirm such
speculation. Moreover, multi-omic single-cell technologies will further improve our understanding
of the regulatory mechanisms of the potential TFs related to lineage-specific H3K9me3
establishment. We have added the statement in the Discussion part of the revised manuscript.

5. Fig. 1B, the K9me3 enrichment is quite low at *Zscan4* and *MERVL* in both CC and fertilized
embryos in 1-2C embryos, contrary to the authors' statements. This should be clarified.

We believe this owes to the rather strong H3K9me3 signals at *MERVL* loci after the 4-cell stage
(Figure R13). When we enlarge this image, we can find the H3K9me3 enrichment in CC, which is
similar to that in 6hpa and 14hpa SCNT embryos

**Figure R13. H3K9me3 signals at *Zscan4d* and *MERVL* loci in fertilized and SCNT embryos before the 2-cell stage.**

6. Figure 1D appeared earlier than Figure 1C.

The reviewer may have missed our statement about “Notably, we observed consistently more
H3K9me3-occupied regions in SCNT embryos at various stages than in fertilized embryos; these
marks may block not only ZGA but also reprogramming at later stages (Figures 1B and 1C).” which
is before figure 1D.

7. Does *KDM4D* overexpression rescue *MERVL* expression defects too in SCNT embryos?

We downloaded the data of SCNT 2C transcriptome after *Kdm4d* OE from a previous study
(GSE59073) (Matoba et al., 2014) (Figure R14). The result supported that expression defects of
*MERVL* and *MT2* were rescued by *Kdm4d* OE.

**Figure R14. *Kdm4d* OE increased the expression level of MERVL and MT2 in SCNT 2-cells.**

**8. Figure 2C, the aphidicolin-treated sample showed only one pronucleus?**

We thought the reviewer mean Figure S2C. Because the pronucleus in SCNT embryos is pseudo-
 pronucleus, the size is not very consistent. The aphidicolin-treated sample shown in Figure S2C
 contains 2 pseudo-pronuclei, with the bigger one at the upper and the smaller one at the lower
 part of the signal region.

**9. Line 358. A citation is missing.**

We appreciate the reviewer and have cited the reference in the revised manuscript.

**10. Lines 370-375. It is unclear what the authors meant by “artificial activation” when referring to
 the *Kdm4a*'s role in MII oocytes.**

SCNT embryos are activated by *in vitro* Sr^{2+} treatment to initiate embryo development, thus we
 call it “artificial activation” to distinguish SCNT from fertilized embryos.

References:

- Burton, A., Brochard, V., Galan, C., Ruiz-Morales, E.R., Rovira, Q., Rodriguez-Terrones, D., Kruse, K., Le Gras, S., Udayakumar, V.S., Chin, H.G., *et al.* (2020). Heterochromatin establishment during early mammalian development is regulated by pericentromeric RNA and characterized by non-repressive H3K9me3. *Nat Cell Biol* 22, 767-778.
- Liu, X., Wang, C., Liu, W., Li, J., Li, C., Kou, X., Chen, J., Zhao, Y., Gao, H., Wang, H., *et al.* (2016). Distinct features of H3K4me3 and H3K27me3 chromatin domains in pre-implantation embryos. *Nature* 537, 558-562.
- Matoba, S., Liu, Y., Lu, F., Iwabuchi, K.A., Shen, L., Inoue, A., and Zhang, Y. (2014). Embryonic development following somatic cell nuclear transfer impeded by persisting histone methylation. *Cell* 159, 884-895.
- Wang, C., Liu, X., Gao, Y., Yang, L., Li, C., Liu, W., Chen, C., Kou, X., Zhao, Y., Chen, J., *et al.* (2018). Reprogramming of H3K9me3-dependent heterochromatin during mammalian embryo development. *Nat Cell Biol* 20, 620-631.
- Xu, R., Li, S., Wu, Q., Li, C., Jiang, M., Guo, L., Chen, M., Yang, L., Dong, X., Wang, H., *et al.* (2022). Stage-specific H3K9me3 occupancy ensures retrotransposon silencing in human pre-implantation embryos. *Cell Stem Cell* 29, 1051-1066 e1058.

REVIEWERS' COMMENTS

Reviewer #1 (Remarks to the Author):

I appreciate the authors for responding to my comments, by conducting the necessary experiments, analyzing them accurately, and making the conclusion carefully. Especially the SCNT experiments using immature Sertoli cells are of great importance. They found that Sertoli cell-derived SCNT embryos showed an inappreciable distinction between ICM and TE on H3K9me3 establishment as cumulus-derived SCNT embryos. However, it became apparent that the Sertoli and cumulus clones were critically different in the involvement of Mcrs1 in epigenetic errors associated with SCNT. Even though this important discovery was made, the revised version does not mention the results of these Sertoli cell clone experiments at all. For example, the abstract only shows the results from the cumulus cell clone, which would lead misunderstanding of the readers; they would consider the involvement of Mcrs1 to be a phenomenon common to all mouse SCNT experiments in general, despite the fact that it is donor-cell specific.

Therefore, I strongly recommend the authors to include this important Sertoli cell clone results in the revised version, adequately describing what are common between the two types of SCNT and what are different between them. Although this would involve rewriting the Abstract and Results, I believe this is essential for an accurate understanding of mouse clones by the researchers in the developmental biology field.

Reviewer #2 (Remarks to the Author):

The authors have appropriately addressed the reviewers' concerns. Thus, I support the publication of this paper in Nature Communications. I believe that the data presented in Figure R1 in the response letter provide valuable insights into the SCNT field and should be included in the final manuscript.

Reviewer #3 (Remarks to the Author):

The authors did an excellent job in revising the paper, which well addressed my previous concerns. I support the publication of this manuscript with some minor comments below.

1. Figure R13, a somewhat zoomed-out snapshot should be provided to show the background H3K9me3 levels in neighbor regions, without which it is difficult to know if H3K9me3 showed by the authors is true signals or background fluctuations.
2. Comment 6. It would be helpful to show the restored H3K9me3 upon Mcrs1 using heatmaps, in addition to boxplots, with appropriate negative controls. This is a rather surprising result and requires careful analyses with the right controls to make sure that the conclusion is correct. In addition, Figure 6I needs to show the original Mcrs1 binding track in addition to the Mcrs1 peak.

Dear Editor,

We appreciate your and the reviewers' time and effort in reviewing our manuscript. We have summarized our response to reviewers and also revised our manuscript accordingly. All the other files and information are prepared according to the Author Checklist.

We have listed the point-to-point response to the comments raised by all three reviewers. We hope that with this sufficient revision, our study would be suitable to be published in Nature Communications. We have highlighted the revised contents in the revised manuscript. The reviewer's comments are in blue followed by our response in black.

Reviewers' Comments to the Authors:

Reviewer #1:

I appreciate the authors for responding to my comments, by conducting the necessary experiments, analyzing them accurately, and making the conclusion carefully. Especially the SCNT experiments using immature Sertoli cells are of great importance. They found that Sertoli cell-derived SCNT embryos showed an inappreciable distinction between ICM and TE on H3K9me3 establishment as cumulus-derived SCNT embryos. However, it became apparent that the Sertoli and cumulus clones were critically different in the involvement of Mcrs1 in epigenetic errors associated with SCNT. Even though this important discovery was made, the revised version does not mention the results of these Sertoli cell clone experiments at all. For example, the abstract only shows the results from the cumulus cell clone, which would lead misunderstanding of the readers; they would consider the involvement of Mcrs1 to be a phenomenon common to all mouse SCNT experiments in general, despite the fact that it is donor-cell specific.

Therefore, I strongly recommend the authors to include this important Sertoli cell clone results in the revised version, adequately describing what are common between the two types of SCNT and what are different between them. Although this would involve rewriting the Abstract and Results, I believe this is essential for an accurate understanding of mouse clones by the researchers in the developmental biology field.

We appreciate the reviewer's suggestion. In the revised manuscript, we have included the results from the Sertoli cell clone.

Reviewer #2:

The authors have appropriately addressed the reviewers' concerns. Thus, I support the publication of this paper in Nature Communications. I believe that the data presented in Figure R1 in the response letter provide valuable insights into the SCNT field and should be included in the final manuscript.

We appreciate the reviewer's positive response to our revision. We have included the data and relative information from Fig. R1 in the revised manuscript.

Reviewer #3 (Remarks to the Author):

The authors did an excellent job in revising the paper, which well addressed my previous concerns. I support the publication of this manuscript with some minor comments below.

1. Figure R13, a somewhat zoomed-out snapshot should be provided to show the background H3K9me3 levels in neighbor regions, without which it is difficult to know if H3K9me3 showed by the authors is true signals or background fluctuations.

We thank the reviewer for pointing this out. We displayed the H3K9me3 signal of the zoomed-out region within 300 kb surrounding *Zscan4d*, and the H3K9me3 signal of whole chromosome 7 where *Zscan4d* is located (Fig. R1). As shown, we observed a significant enrichment of H3K9me3 on the SCNT in the locus of *Zscan4d* and *MERVL*, compared to the background signal in other regions.

Fig. R1 | H3K9me3 signals at *Zscan4d* and *MERVL* loci in fertilized and SCNT embryos before the 2-cell stage. (a) 300 kb zoomed-out. (b) Whole chromosome 7.

2. Comment 6. It would be helpful to show the restored H3K9me3 upon *Mcrs1* using heatmaps, in addition to boxplots, with appropriate negative controls. This is a rather surprising result and requires careful analyses with the right controls to make sure that the conclusion is correct. In addition, Figure 6I needs to show the original *Mcrs1* binding track in addition to the *Mcrs1* peak.

We apologize for providing the wrong figure number in Comment 6. In fact, we presented a heatmap of changes in H3K9me3 on *Mcrs1* binding peaks and found that approximately 33% of H3K9me3 were successfully corrected by *Mcrs1* OE (Fig. R2a). The majority of the uncorrected regions were used as negative controls. This indicates that only a small portion of H3K9me3 can be rescued by *Mcrs1* overexpression, but these regions have important regulatory functions (Fig.R2b). Additionally, we have added a *Mcrs1* binding signal track from the public database in the revised Fig.6I (Fig. R2c).

Fig.R2 | *Mcrs1* OE rescued H3K9me3 levels on the binding sites of *Mcrs1* in ICM and TE.

(a) Heat map showing the scaled H3K9me3 signals in ICM and TE of fertilized and SCNT (control and *Mcrs1* OE) embryos. The regions were classified the same as in Fig. 4b. (b) Go-term analysis of genes marked by *Mcrs1* OE-rescued and un-rescued H3K9me3. (c) Genome Browser view of H3K9me3 signals around the *Mcrs1* binding sites in the fertilized (green), SCNT (red), *Mcrs1* OE SCNT (orange) embryos, and the *Mcrs1* binding track (grey, published in GSE51746). The signals are presented as the RPKM.